# SOMBRL: Scalable and Optimistic Model-Based RL

**Bhavya Sukhija**
Department of Computer Science
ETH Zurich
sukhijab@ethz.ch

**Lenart Treven**
Department of Computer Science
ETH Zurich
trevenl@ethz.ch

**Carmelo Sferrazza**
Berkeley AI Research
UC Berkeley
csferrazza@berkeley.edu

**Florian Dörfler**
Department of Electrical Engineering
ETH Zurich
dorfler@ethz.ch

**Pieter Abbeel**
Berkeley AI Research
UC Berkeley
pabbeel@berekeley.edu

**Andreas Krause**
Department of Computer Science
ETH Zurich
krausea@ethz.ch

## Abstract

We address the challenge of efficient exploration in model-based reinforcement learning (MBRL), where the system dynamics are unknown and the RL agent must learn directly from online interactions. We propose **S**calable and **O**ptimistic **MBRL** (SOMBRL), an approach based on the principle of optimism in the face of uncertainty. SOMBRL learns an uncertainty-aware dynamics model and *greedily* maximizes a weighted sum of the extrinsic reward and the agent's epistemic uncertainty. SOMBRL is compatible with any policy optimizers or planners, and under common regularity assumptions on the system, we show that SOMBRL has sublinear regret for nonlinear dynamics in the (*i*) finite-horizon, (*ii*) discounted infinite-horizon, and (*iii*) non-episodic settings. Additionally, SOMBRL offers a flexible and scalable solution for principled exploration. We evaluate SOMBRL on state-based and visual-control environments, where it displays strong performance across all tasks and baselines. We also evaluate SOMBRL on a dynamic RC car hardware and show SOMBRL outperforms the state-of-the-art, illustrating the benefits of principled exploration for MBRL.

## 1   Introduction

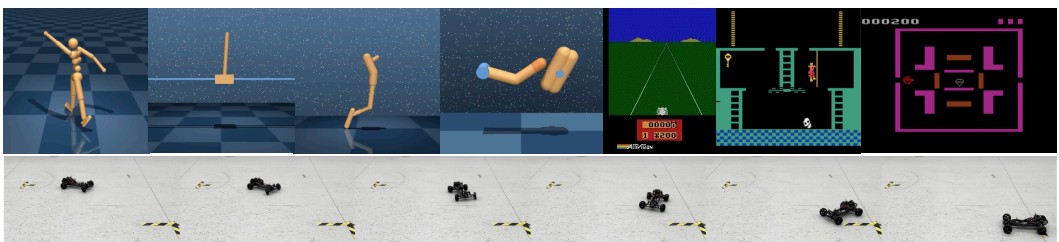

Figure 1: *Top:* We showcase scalability of SOMBRL on visual control tasks from DMC and Atari. *Bottom:* We evaluate SOMBRL on a highly dynamic RC car where we learn to perform a complex parking maneuver in only 20 real-world episodes.

Reinforcement learning (RL) has been successfully applied to a variety of sequential-decision making problems like games (Silver et al., 2017), robotics (Kober et al., 2013; Tang et al., 2025), mobile

39th Conference on Neural Information Processing Systems (NeurIPS 2025).

health interventions (Yom-Tov et al., 2017; Liao et al., 2020), and fine-tuning of large language models (Ouyang et al., 2022). RL offers a flexible learning paradigm, enabling agents to learn directly by interacting with their environment. However, this potential is often not fully realized in practice, as most widely used RL methods (Schulman et al., 2017) are highly sample-inefficient. This mostly rules out their direct application to real-world settings where data is scarce or expensive to acquire.

Model-based RL approaches (Moerland et al., 2023) offer a more sample-efficient alternative and have been successfully used for learning directly in the real-world (Hansen et al., 2022; Wu et al., 2023; Rothfuss et al., 2024). However, these methods are mostly based on naive exploration strategies, such as Boltzmann exploration, which are provably sub-optimal (Cesa-Bianchi et al., 2017) and often struggle in the presence of sparse rewards.

**Related Works** Several works study principled exploration approaches in RL (Even-Dar & Mansour (2001); Jaksch et al. (2010); Abbasi-Yadkori & Szepesvári (2011); Cohen et al. (2019); Dean et al. (2020); Kakade et al. (2020); Curi et al. (2020); Neu & Pike-Burke (2020); Foster & Rakhlin (2023); Wagenmaker et al. (2023); Sukhija et al. (2024c), see Section 3 and Appendix A for more details). In particular, optimism in the face of uncertainty is a celebrated exploration principle with strong theoretical guarantees for model-based RL (Brafman & Tennenholtz, 2002; Jaksch et al., 2010; Kakade et al., 2020; Curi et al., 2020; Moulin & Neu, 2023; Sukhija et al., 2024b). However, in practice, these algorithms are computationally prohibitive. As a result, naive exploration techniques remain dominant in real-world applications due to their simplicity. We address this gap between theory and practice and propose a simple yet principled method, SOMBRL, for exploration that enjoys convergence guarantees across several RL settings. This is in contrast to prior works, which design and study algorithms only for specific settings, e.g., Curi et al. (2020); Kakade & Langford (2002) study the finite-horizon setting, Sukhija et al. (2024c) the unsupervised RL setting, and Sukhija et al. (2024b) the nonepisodic one. In addition to the theoretical guarantees, SOMBRL is also much more scalable and, unlike the aforementioned works, can be applied to high-dimensional settings such as visual control problems and real-world hardware platforms. We demonstrate this in our experiments (Section 6).

The differences between SOMBRL and prior work on exploration in MBRL are summarized in Table 1, and we provide a more detailed discussion of related works in Appendix A.

**Contributions**

1. We propose SOMBRL, a principled yet efficient exploration strategy for model-based RL. SOMBRL is based on the principle of optimism in the face of uncertainty, and *greedily* maximizes a weighted sum of the extrinsic reward and the agent's epistemic uncertainty/disagreement. Therefore, the agent selects policies that maximize rewards while also exploring less visited areas of the state space where the uncertainty about the system is high.

2. We show, under common regularity assumptions on the dynamics, that combining extrinsic rewards with the agent's epistemic uncertainty gives anytime high probability value-function bounds, which could be of independent interest to applications such as safe RL (Brunke et al., 2022) and offline RL (Levine et al., 2020). We leverage this key insight and show that SOMBRL has sublinear regret for finite-horizon, discounted infinite-horizon, and nonepisodic settings with continuous state and action spaces. Our regret bounds are comparable to the ones derived by prior work (Kakade et al., 2020; Curi et al., 2020; Sukhija et al., 2024b), but our algorithm is considerably simpler and more scalable.

3. We validate SOMBRL on standard deep RL benchmarks, showing that it outperforms several naive exploration baselines and scales effectively to high-dimensional tasks, such as visual control. We also *evaluate* SOMBRL *in the real-world* on a dynamic RC car (see Figure 1), where it learns an agile parking maneuver in only 20 trials, outperforming the state-of-the-art (Rothfuss et al., 2024) w.r.t. performance and sample efficiency. To the best of our knowledge, this is the first empirical demonstration of optimistic exploration in model-based deep RL for high-dimensional and real-world settings.

## 2 Problem Setting

We consider a discrete-time dynamical system of the form $\boldsymbol{x}_{t+1} = \boldsymbol{f}^*(\boldsymbol{x}_t, \boldsymbol{u}_t) + \boldsymbol{w}_t$, where $\boldsymbol{x}_t \in \mathcal{X} \subseteq \mathbb{R}^{d_x}$ is the state, $\boldsymbol{u}_t \in \mathcal{U} \subseteq \mathbb{R}^{d_u}$ the control input, and $\boldsymbol{w}_t \in \mathcal{W} \subseteq \mathbb{R}^{\boldsymbol{w}}$ the process noise[1]. The dynamics $\boldsymbol{f}^*$ are unknown.

---

[1] For our theory, we assume the process noise to be known, but our algorithm can learn it from data.

Table 1: Comparison of SOMBRL and prior work on model-based RL (MBRL) in terms of sublinear regret/sample complexity for the kernelized settings and scalability. Kakade et al. (2020) give regret bound for the finite-horizon setting with Gaussian noise and Curi et al. (2020) for the sub-Gaussian noise case. We show that SOMBRL has sublinear regret for both settings (Theorem 5.4 and Theorem B.18)

| | Finite Horizon | $\gamma$-discounted infinite-horizon | Nonepisodic | Unsupervised RL | Scalable + Practical |
|---|---|---|---|---|---|
| Greedy, e.g., Mean planning or Deisenroth & Rasmussen (2011); Chua et al. (2018) | ✗ | ✗ | ✗ | ✗ | ✓ |
| Curi et al. (2020); Kakade et al. (2020) | ✓ | ✗ | ✗ | ✗ | ✗ |
| Sukhija et al. (2024b) | ✗ | ✗ | ✓ | ✗ | ✗ |
| Sukhija et al. (2024c) | ✗ | ✗ | ✗ | ✓ | ✗ |
| **SOMBRL (ours)** | ✓ | ✓ | ✓ | ✓ | ✓ |

**Task** In the finite-horizon RL setting (Puterman, 2014), we are given a reward function $r : \mathcal{X} \times \mathcal{U} \to \mathbb{R}$, and want to learn a policy that maximizes the following objective

$$J(\boldsymbol{\pi}^*) = \max_{\boldsymbol{\pi} \in \Pi} J(\boldsymbol{\pi}) = \max_{\boldsymbol{\pi} \in \Pi} \mathbb{E}_{\boldsymbol{\pi}} \left[ \sum_{t=0}^{T-1} r(\boldsymbol{x}_t, \boldsymbol{u}_t) \right], \tag{1}$$

where action $\boldsymbol{u}_t$ follows policy $\boldsymbol{\pi}$, i.e., $\boldsymbol{u}_t \sim \boldsymbol{\pi}(\boldsymbol{x}_t)$. Moreover, we consider the episodic RL setting, with episodes $n \in \{1, \ldots, N\}$, and study a model-based approach. Accordingly, at the beginning of episode $n$, we select and roll out a policy $\boldsymbol{\pi}_n$ for $T$ steps on the true system. We then use the data collected from the rollouts to estimate the true dynamics $\boldsymbol{f}^*$. The goal is to find a policy that performs as well as $\boldsymbol{\pi}^*$, as quickly as possible. Therefore a natural performance metric in this context is the *cumulative regret* $R_N = \sum_{n=1}^{N} J(\boldsymbol{\pi}^*) - J(\boldsymbol{\pi}_n)$. In the following sections, we show that our proposed algorithm achieves sublinear regret. While in the main text for clarity we focus on the finite-horizon episodic setting, in Section 5, we show that our approach has sublinear regret also for

$\gamma$-discounted infinite-horizon, episodic:      and average reward, nonepisodic settings:

$$J_\gamma(\boldsymbol{\pi}^*) = \max_{\boldsymbol{\pi} \in \Pi} \mathbb{E}_{\boldsymbol{\pi}} \left[ \sum_{t=0}^{\infty} \gamma^t r(\boldsymbol{x}_t, \boldsymbol{u}_t) \right] \quad (2) \quad J_{\text{avg}}(\boldsymbol{\pi}^*) = \max_{\boldsymbol{\pi} \in \Pi} \limsup_{T \to \infty} \frac{1}{T} \mathbb{E}_{\boldsymbol{\pi}} \left[ \sum_{t=0}^{T-1} r(\boldsymbol{x}_t, \boldsymbol{u}_t) \right] \quad (3)$$

## 3 Exploration Strategies in MBRL

In MBRL, we learn a model of the true dynamics $\boldsymbol{f}^*$ and use our learned model to select/update the next policy for data acquisition. Exploration algorithms for MBRL determine how the policy should be chosen given our learned model. Common strategies for this choice are (*i*) greedy planning, (*ii*) Thompson sampling, and (*iii*) optimistic exploration. We discuss these in detail below.

Let $J(\boldsymbol{\pi}, \boldsymbol{f})$ be the the expected returns under the policy $\boldsymbol{\pi}$ and dynamics $\boldsymbol{f}$, that is

$$J(\boldsymbol{\pi}, \boldsymbol{f}) = \mathbb{E}_{\boldsymbol{\pi}} \left[ \sum_{t=0}^{T-1} r(\boldsymbol{x}'_t, \boldsymbol{u}_t) \right], \quad \boldsymbol{x}'_{t+1} = \boldsymbol{f}(\boldsymbol{x}'_t, \boldsymbol{u}_t) + \boldsymbol{w}_t, \boldsymbol{x}'_0 = \boldsymbol{x}_0,$$

and $\boldsymbol{\mu}_n$ our mean estimate of the dynamics $\boldsymbol{f}^*$ at episode $n$.

**Greedy planning** The simplest selection strategy is to pick the policy $\boldsymbol{\pi}_n$ that maximizes the expected returns for our estimated dynamics $\boldsymbol{\mu}_n$.

$$\boldsymbol{\pi}_n^{\text{MEAN}} = \arg\max_{\boldsymbol{\pi} \in \Pi} J(\boldsymbol{\pi}, \boldsymbol{\mu}_n) \tag{4}$$

This strategy is greedy as it does not directly encourage exploration in areas where we have limited data or where our model has high uncertainty. Instead, it exploits our estimate $\boldsymbol{\mu}_n$ of the dynamics. This is the basis of methods such as those of Janner et al. (2019); Hafner et al. (2023), where exploration is induced using a stochastic policy that is optimized with an entropy bonus.

To incorporate epistemic uncertainty in our learned model and avoid overfitting to misestimated dynamics, Deisenroth & Rasmussen (2011); Chua et al. (2018); Rothfuss et al. (2024) learn a Bayesian model of $\boldsymbol{f}^*$: $p(\boldsymbol{f}|\mathcal{D}_{1:n})$. Here $\mathcal{D}_{1:n} = \cup_{i \leq n} \mathcal{D}_i$, and $\mathcal{D}_i = \{(\boldsymbol{x}_{t,i}, \boldsymbol{u}_{t,i}, \boldsymbol{x}_{t+1,i})\}_{t=0}^{T-1}$ is the data collected in episode $i$. The policy $\boldsymbol{\pi}_n$ is then selected as

$$\boldsymbol{\pi}_n^{\text{GREEDY}} = \arg\max_{\boldsymbol{\pi} \in \Pi} \mathbb{E}_{\boldsymbol{f} \sim p(\boldsymbol{f}|\mathcal{D}_{1:n})}[J(\boldsymbol{\pi}, \boldsymbol{f})]. \tag{5}$$

Curi et al. (2020) show that greedy planning may fail to perform well in practice, especially for difficult exploration problems (e.g., in context of action penalties).

**Thompson Sampling** In Thompson sampling (TS), we also learn a Bayesian model $p(\boldsymbol{f}|\mathcal{D}_{1:n})$ and pick policies by maximizing the reward under $\boldsymbol{f}$ sampled from the posterior

$$\boldsymbol{\pi}_n^{\mathrm{TS}} = \underset{\boldsymbol{\pi} \in \Pi}{\arg\max} \ J(\boldsymbol{\pi}, \boldsymbol{f}), \ \boldsymbol{f} \sim p(\boldsymbol{f}|\mathcal{D}_{1:n}). \tag{6}$$

While TS encourages exploration in a theoretically grounded manner (Russo et al., 2018), in practice, it is often intractable to sample a function $\boldsymbol{f}$ from $p(\boldsymbol{f}|\mathcal{D}_{1:n})$.

**Optimistic Exploration** This strategy is based on the principle of optimism in the face of uncertainty. Optimistic exploration approaches maintain a set of *plausible dynamics models* $\mathcal{M}_n$ at each episode $n$, e.g., the set of functions that have a high probability w.r.t. a learned Bayesian model $p(\boldsymbol{f}|\mathcal{D}_{1:n})$. The policy is then selected according to

$$\boldsymbol{\pi}_n^{\mathrm{OE}} = \underset{\boldsymbol{\pi} \in \Pi, \boldsymbol{f} \in \mathcal{M}_n}{\arg\max} \ J(\boldsymbol{\pi}, \boldsymbol{f}) \tag{7}$$

There are several works that study optimistic exploration theoretically (Jaksch et al., 2010; Kakade et al., 2020; Curi et al., 2020; Treven et al., 2024; Sukhija et al., 2024b). However, optimizing $\boldsymbol{f}$ over $\mathcal{M}_n$, typically a difficult non-convex constraint, is often computationally prohibitive, restricting the application of these methods to fairly low-dimensional settings. The most efficient solvers of the optimization problem (7), to the best of our knowledge, are based on a reparametrization trick which introduces additional hallucinated controls (Curi et al., 2020). This increases the total control dimension from $d_{\boldsymbol{u}}$ to $d_{\boldsymbol{u}} + d_{\boldsymbol{x}}$, which is prohibitive in high-dimensional domains.

## 4 SOMBRL: Scalable and Optimistic MBRL

We now present SOMBRL, our approach for efficient optimistic exploration in MBRL, which alternates between two steps. First, given a dataset of transitions $\mathcal{D}_{1:n}$, we learn an uncertainty-aware model of the unknown dynamics $\boldsymbol{f}^*$. That is, after each episode $n$, we learn a mean estimate $\boldsymbol{\mu}_n$ of $\boldsymbol{f}^*$ and quantify our epistemic uncertainty $\boldsymbol{\sigma}_n$ over the estimate. Models such as Gaussian processes (GPs) (Rasmussen & Williams, 2005) can be directly used for this purpose. Bayesian deep learning approaches such as deep ensembles are also commonly used to quantify epistemic uncertainty or model disagreement in RL (Chua et al., 2018; Pathak et al., 2019; Curi et al., 2020; Sekar et al., 2020; Sukhija et al., 2024c). In the second step, we solve the following optimization problem for the policy $\boldsymbol{\pi}_n$

$$\boldsymbol{\pi}_n := \underset{\boldsymbol{\pi} \in \Pi}{\arg\max} \ \underbrace{\mathbb{E}_{\boldsymbol{\pi}}\left[\sum_{t=0}^{T-1} r(\boldsymbol{x}_t', \boldsymbol{u}_t) + \lambda_n \|\boldsymbol{\sigma}_n(\boldsymbol{x}_t', \boldsymbol{u}_t)\|\right]}_{J_n(\boldsymbol{\pi})}, \ \boldsymbol{x}_{t+1}' = \boldsymbol{\mu}_n(\boldsymbol{x}_t', \boldsymbol{u}_t) + \boldsymbol{w}_t, \tag{8}$$

where $\lambda_n$ is a positive constant which is used to trade off maximizing the extrinsic reward and model uncertainty (see Appendix B for how $\lambda_n$ is defined in theory and Section 5.2 and Appendix D for how it is selected empirically). Note that in Equation (8), we use the mean dynamics for planning and only use the epistemic uncertainty as an additional *intrinsic* reward. Compared to the principled exploration strategies from Section 3, our approach does not require sampling from or maximizing over the dynamics. This makes SOMBRL much simpler and more scalable. Moreover, SOMBRL can be combined with any model-based algorithm such as those of Janner et al. (2019); Hafner et al. (2023); Rothfuss et al. (2024). The only additional modification we make to these methods is that we add the epistemic uncertainty to the extrinsic reward. Also note that without the epistemic uncertainty reward, i.e., $\lambda_n = 0$, the agent follows the greedy strategy discussed in Section 3 and for $\lambda \to \infty$ the agent performs unsupervised exploration (Pathak et al., 2017; Sekar et al., 2020; Buisson-Fenet et al., 2020; Sukhija et al., 2024c). Therefore, we use the model uncertainty to facilitate principled exploration for the agent.

In the following, we show that by optimizing our objective in Equation (8), we are effectively maximizing an optimistic estimate of $J(\boldsymbol{\pi}^*)$, i.e., we are also performing optimistic exploration. Accordingly, our approach enjoys the same guarantees as other optimistic MBRL algorithms but is much simpler, computationally cheaper, and scalable to high-dimensional settings.

## 5 Theoretical Results

For our analysis, we make some common assumptions on the underlying dynamics $\boldsymbol{f}^*$.

## 5.1 Assumptions

We first make continuity assumptions on the system. These assumptions are common in the control theory (Khalil, 2015) and reinforcement learning literature (Curi et al., 2020; Sussex et al., 2023; Sukhija et al., 2024c).

**Assumption 5.1** (Continuous closed-loop dynamics, bounded rewards, and Gaussian noise.). The dynamics model $\boldsymbol{f}^*$ and all $\boldsymbol{\pi} \in \Pi$ are continuous. Furthermore, we assume that the reward is bounded, i.e., $r : \mathcal{X} \times \mathcal{U} \to [0, R_{\max}]$, and process noise is i.i.d. Gaussian[2] with variance $\sigma^2$, i.e., $\boldsymbol{w}_t \overset{i.i.d}{\sim} \mathcal{N}(\boldsymbol{0}, \sigma^2 \boldsymbol{I})$.

SOMBRL learns an uncertainty-aware model of the true dynamics $\boldsymbol{f}^*$, i.e., a mean $\boldsymbol{\mu}_n$ and uncertainty $\boldsymbol{\sigma}_n$ estimate. In our theoretical analysis, we focus on Gaussian Process (GP) dynamics models. For GPs, $\boldsymbol{\mu}_n$ and $\boldsymbol{\sigma}_n$ have a closed-form solution, and our learned model is calibrated (c.f.,Rothfuss et al. (2023) or Appendix B.1 for the definition of well-calibrated models or Kuleshov et al. (2018) on calibration for BNNs). Generally, our guarantees can be extended to broader classes of well-calibrated models, e.g., BNNs (similar to Curi et al. (2020)).

We assume that $\boldsymbol{f}^*$ resides in a Reproducing Kernel Hilbert Space (RKHS) of vector-valued functions.

**Assumption 5.2.** We assume that the functions $f_j^*$, $j \in \{1, \ldots, d_{\boldsymbol{x}}\}$ lie in a RKHS with kernel $k$ and have a bounded norm $B$, that is $\boldsymbol{f}^* \in \mathcal{H}_{k,B}^{d_{\boldsymbol{x}}}$, with $\mathcal{H}_{k,B}^{d_{\boldsymbol{x}}} = \{\boldsymbol{f} \mid \|f_j\|_k \leq B, j = 1, \ldots, d_{\boldsymbol{x}}\}$. Moreover, we assume that $k(\boldsymbol{z}, \boldsymbol{z}) \leq \sigma_{\max}$ for all $\boldsymbol{x} \in \mathcal{X}$.

Assumption 5.2 ensures that we can model and learn $\boldsymbol{f}^*$ with GPs. This assumption is common in the Bayesian optimization (Srinivas et al., 2012; Chowdhury & Gopalan, 2017) and RL literature (Kakade et al., 2020; Curi et al., 2020). Moreover, GPs are nonparametric models and can learn very complex classes of nonlinear functions (Rasmussen & Williams, 2005).

The posterior mean $\boldsymbol{\mu}_n(\boldsymbol{z}) = [\mu_{n,j}(\boldsymbol{z})]_{j \leq d_{\boldsymbol{x}}}$ and epistemic uncertainty $\boldsymbol{\sigma}_n(\boldsymbol{z}) = [\sigma_{n,j}(\boldsymbol{z})]_{j \leq d_{\boldsymbol{x}}}$ can then be obtained using the following formula

$$
\begin{aligned}
\mu_{n,j}(\boldsymbol{z}) &= \boldsymbol{k}_n^\top(\boldsymbol{z})(\boldsymbol{K}_n + \sigma^2 \boldsymbol{I})^{-1} \boldsymbol{y}_{1:n}^j, \\
\sigma_{n,j}^2(\boldsymbol{z}) &= k(\boldsymbol{z}, \boldsymbol{z}) - \boldsymbol{k}_n^\top(\boldsymbol{z})(\boldsymbol{K}_n + \sigma^2 \boldsymbol{I})^{-1} \boldsymbol{k}_n(\boldsymbol{z}),
\end{aligned}
\tag{9}
$$

Here, $\boldsymbol{y}_{1:n}^j$ corresponds to the noisy measurements of $f_j^*$, i.e., the observed next state from the transitions dataset $\mathcal{D}_{1:n}$, $\boldsymbol{k}_n(\boldsymbol{z}) = [k(\boldsymbol{z}, \boldsymbol{z}_i)]_{\boldsymbol{z}_i \in \mathcal{D}_{1:n}}$, and $\boldsymbol{K}_n = [k(\boldsymbol{z}_i, \boldsymbol{z}_l)]_{\boldsymbol{z}_i, \boldsymbol{z}_l \in \mathcal{D}_{1:n}}$ is the data kernel matrix. The restriction on the kernel $k(\boldsymbol{z}, \boldsymbol{z}) \leq \sigma_{\max}$ implies boundedness of $\boldsymbol{f}^*$ and has also appeared in works studying the episodic setting for nonlinear dynamics (Mania et al., 2020; Kakade et al., 2020; Curi et al., 2020; Wagenmaker et al., 2023; Sukhija et al., 2024c). We can also define $\boldsymbol{f}^*$ such that $\boldsymbol{x}_k = \boldsymbol{x}_{k-1} + \boldsymbol{f}^*(\boldsymbol{x}_{k-1}, \boldsymbol{u}_{k-1}) + \boldsymbol{w}_{k-1}$ in which case the boundedness of $\boldsymbol{f}^*$ captures many real-world systems.

Our theoretical results depend on the *maximum information gain* of kernel $k$ (Srinivas et al., 2012), defined as

$$
\Gamma_N(k) = \max_{\mathcal{A} \subset \mathcal{X} \times \mathcal{U}; |\mathcal{A}| \leq N} \frac{1}{2} \log \left| \boldsymbol{I} + \sigma^{-2} \boldsymbol{K}_N \right|.
$$

$\Gamma_N$ is a measure of the complexity for learning $\boldsymbol{f}^*$ from $N$ episodes and is sublinear for many kernels (e.g., $\mathcal{O}(\log^{d_x + d_u + 1}(N))$ for the squared exponential (RBF) kernel, $\mathcal{O}((d_x + d_u) \log(N))$ for the linear kernel). In Appendix B, we report the dependence of $\Gamma_N$ on $N$ in Table 2.

Next, we present the following Lemma, which states that $J_n(\boldsymbol{\pi}_n)$ from Equation (8) is an optimistic estimate of $J(\boldsymbol{\pi}^*)$.

**Lemma 5.3.** *Let Assumption 5.1 and Assumption 5.2 hold. Then, there exists a $\lambda_n \in \Theta(\sqrt{\Gamma_N})$, such that we have $\forall n > 0$, $\boldsymbol{\pi} \in \Pi$, with probability at least $1 - \delta$, that $J(\boldsymbol{\pi}) \leq J_n(\boldsymbol{\pi})$. Moreover, we have $J(\boldsymbol{\pi}^*) \leq J_n(\boldsymbol{\pi}_n)$.*

*Proof Sketch*: Crucially, we leverage the policy difference lemma (Kakade & Langford, 2002), which bounds the difference between the performance of two different policies with the advantage function. However, we study this lemma for a fixed policy but different dynamics, effectively obtaining a simulation lemma (Kearns & Singh, 2002) for our setting. Next, we bound the difference

---

[2]For clarity of exposition, we focus on the setting with Gaussian noise. In Appendix B, we also perform the analysis for the more general sub-Gaussian noise case.

in performance between the mean dynamics and the true one and show that this is proportional to the model epistemic uncertainty.

Lemma 5.3 shows that for all policies $\boldsymbol{\pi} \in \Pi$, $J_n(\boldsymbol{\pi})$ gives an upper bound on the true return $J(\boldsymbol{\pi})$. This result is of independent interest and can be applied to settings beyond online RL, such as safe RL (Brunke et al., 2022; As et al., 2024) and offline RL (Levine et al., 2020; Yu et al., 2020; Rigter et al., 2022). The exact bound for $\lambda_n$ is provided in Lemma B.3 in Appendix B.

Finally, we present our main theorem, which bounds the regret of SOMBRL.

**Theorem 5.4** (Finite horizon setting). *Let Assumption 5.1 and Assumption 5.2 hold. Then we have* $\forall N > 0$ *with probability at least* $1 - \delta$, $R_N \leq \mathcal{O}\left(\Gamma_N^{3/2}\sqrt{N}\right)$.

*Proof Sketch*: To bound the regret, we first prove that Equation (8) is an optimistic estimate of Equation (1). Then, we use Lemma 5.3 to bound the difference in performance for extrinsic rewards between the mean and true dynamics with the epistemic uncertainty of the collected rollout. Next, we analyse the intrinsic reward term and also show that it's bounded by the epistemic uncertainty of the collected rollout. This allows us to relate the cumulative regret $R_N$ with the information gain $\Gamma_N$ and obtain the final bound.

Theorem 5.4 guarantees sublinear regret for a rich class of RKHS functions. Accordingly, for many RKHS, our algorithm enjoys the same asymptotic guarantees as Kakade et al. (2020). Note that the regret bound from Kakade et al. (2020) is an order of $\sqrt{\Gamma_N}$ better. On the other hand, SOMBRL is a much simpler and more scalable algorithm. In Appendix B, we show that SOMBRL improves the regret bound from Curi et al. (2020), for the sub-Gaussian noise case, by an exponential factor of $\Gamma_N^T$. Below, we also provide our regret bounds for the $\gamma$-discounted and the non-episodic setting[3].

**Theorem 5.5** ($\gamma$-discounted, infinite horizon setting). *Let* $R_N = \sum_{n=1}^{N} J_\gamma(\boldsymbol{\pi}^*) - J_\gamma(\boldsymbol{\pi}_n)$. *Under the Assumption 5.1 and Assumption 5.2, we have for the $\gamma$-discounted infinite (Equation (2)) horizon setting* $\forall N > 0$ *that with probability at least* $1 - \delta$, $R_N \leq \mathcal{O}\left(\Gamma_{N\log(N)}^{3/2}\sqrt{N}\right)$.

*Proof Sketch*: The regret decomposition for this setting is similar to the finite-horizon case (Theorem 5.4). However, in contrast to the finite-horizon case, where each episode has a fixed length $T$, we care about the infinite horizon in the $\gamma$-discounted case. Therefore, to obtain sublinear regret for this setting, we require the agent to observe the system for longer horizons, i.e., the horizon $T(n) \to \infty$ for $n \to \infty$. We ensure this by picking $T(n) \in \Theta(\log(n))$ and show that this is sufficient to obtain sublinear regret.

In Theorem 5.5, we show that even though we truncate each episode after $T(n)$ steps, SOMBRL has sublinear regret w.r.t. the infinite horizon objective. Moreover, the regret for this setting follows the same structure as for the finite horizon case. To the best of our knowledge, we are the first to give a regret bound for optimistic model-based RL algorithms for the $\gamma$-discounted setting.

Finally, we give our regret bound for the non-episodic setting. As pointed out in Kakade (2003); Sharma et al. (2021), this is the most challenging and closest setting for learning directly in the real-world. Sukhija et al. (2024b) show that optimistic exploration methods have sublinear regret for the nonepisodic setting. However, their proposed algorithm is intractable in practice.

**Theorem 5.6** (Informal statement; nonepisodic average reward case). *Let* $R_N = \sum_{n=1}^{N} \mathbb{E}[J_{avg}(\boldsymbol{\pi}^*) - r(\boldsymbol{x}_n, \boldsymbol{\pi}_n(\boldsymbol{x}_n))]$. *Under the same assumptions as Sukhija et al. (2024b), we have for the average reward setting (Equation (3))* $\forall N > 0$ *that with probability at least* $1 - \delta$, $R_N \leq \mathcal{O}\left(\Gamma_N^{3/2}\sqrt{N}\right)$.

*Proof Sketch*: The regret analysis for this setting is, in spirit, similar to the finite horizon case. However, in the nonepisodic setting, we cannot reset the agent and have to learn from a single trajectory. Therefore, unlike the episodic case, where we update our model and policy after every episode, in the nonepisodic setting we have to decide when to update the agent. For SOMBRL we show that if we only update once we have accumulated enough information, i.e., $\sum_{t=0}^{T(n)-1} \|\boldsymbol{\sigma}_n(\boldsymbol{x}_{t,n}, \boldsymbol{\pi}_n(\boldsymbol{x}_{t,n}))\| > C$, for a positive constant $C$, then SOMBRL has sublinear regret. Intuitively, the model uncertainty measures how much information our agent has acquired since its last update at time step $T(n-1)$. We only update the agent once the information exceeds the threshold $C$.

---

[3]Both Kakade et al. (2020) and Curi et al. (2020) do not provide a regret bound for these settings.

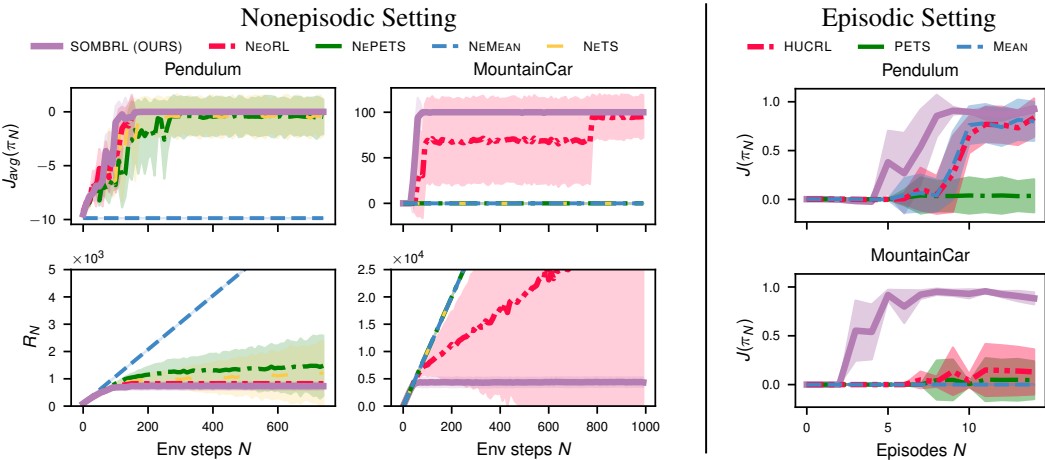

Figure 2: *Left:* Learning curves for the nonepisodic setting with GP dynamics. We report the average reward $J_{avg}(\boldsymbol{\pi}_N)$ and regret $R_N$. The curves are reported with 5 seeds, and we plot the median return with its standard deviation. *Right:* Learning curves for the episodic setting with GP dynamics. We report the median episode reward $J(\boldsymbol{\pi}_N)$ over an episode with 5 seeds and its standard deviation.

In contrast to Sukhija et al. (2024b), SOMBRL is much more tractable, and in Theorem 5.6 we show that SOMBRL also has sublinear regret in the nonepisodic setting and therefore offers a theoretically strong and practical alternative for model-based exploration for this case.

In this section, we have shown SOMBRL, which maximizes a combination of the extrinsic reward and the model epistemic uncertainty, enjoys sublinear regret for common kernels and RL settings. There are principled exploration algorithms designed individually for these settings, e.g., (Kakade et al., 2020; Curi et al., 2020) for the finite-horizon case and Sukhija et al. (2024b) for the non-episodic case. However, they are often intractable/computationally prohibitive. In contrast, SOMBRL works across the different RL settings while also being more practical and scalable.

We present additional theoretical results, for example, a sample complexity bound for unsupervised RL algorithms such as Sekar et al. (2020); Buisson-Fenet et al. (2020); Sukhija et al. (2024c) and a regret bound for the sub-Gaussian noise setting in Appendix B. Our detailed proofs are also provided in Appendix B.

## 5.2   Selecting $\lambda_n$ in practice

The parameter $\lambda_n$ controls the exploration-exploitation trade-off for SOMBRL. In Appendix B we provide the theoretical bound for $\lambda_n$, however in practice, $\lambda_n$ is treated as a hyperparameter. This is similar to other optimistic exploration and intrinsic exploration algorithms (Burda et al., 2018; Kakade et al., 2020; Curi et al., 2020), which also heuristically select the amount of exploration. Sukhija et al. (2024a) empirically study combining extrinsic and intrinsic rewards for model-free algorithms and propose an approach for automatically tuning the intrinsic reward coefficient, i.e., $\lambda_n$. We find their approach works well for our state-based and visual control tasks. Moreover, we describe their approach and how we choose $\lambda_n$ for our experiments in Appendix D.

## 5.3   Application of SOMBRL with GP dynamics

Finally, we empirically validate our theoretical findings for the GP case in Figure 2, where we compare SOMBRL to HUCRL (Curi et al., 2020), PETS (Chua et al., 2018), and greedy (mean) planning in the episodic setting. For the nonepisodic setting, we consider their nonepisodic counterparts as proposed in Sukhija et al. (2024b). We evaluate the algorithms on the Pendulum and MountainCar tasks from the OpenAI Gym benchmark (Brockman et al., 2016). From the experiments, we conclude that SOMBRL performs the best across all baselines for both the episodic and the nonepisodic setting. Moreover, while HUCRL and NEoRL, which explore according to Equation (7), perform better than other baselines, they are worse than SOMBRL. We believe this is because of the practical challenges associated with solving the optimization problem in Equation (7). Moreover, solving Equation (7) is also computationally more expensive. For instance, in our experiments, HUCRL requires roughly $3\times$ more compute time than OMBRL (see Appendix D.5).

# 6 Experiments

In our experiments, we showcase the flexibility and scalability of SOMBRL by combining it with three different model-based RL algorithms; (*i*) MBPO (Janner et al., 2019) for state-based tasks, DREAMER (Hafner et al., 2023) for visual control tasks, and SIMFSVGD (Rothfuss et al., 2024) for our hardware experiment on the RC car. Note that principled exploration methods such as Kakade et al. (2020); Curi et al. (2020) do not scale to the settings, such as visual control tasks, considered in this work. We consider the DeepMind control (DMC) benchmark (Tassa et al., 2018) for the state-based and visual control tasks and test on environments with varying dimensionality[4]. We also evaluate on several environments from the Atari benchmark (Bellemare et al., 2013) for the visual control tasks. In all our experiments, we report the episodic returns using the median over 5 seeds along with its standard deviation. We provide additional experiment details in Appendix D.

**State-based experiments**  We refer to the MBPO version of SOMBRL as MBPO-OPTIMISTIC. The resulting algorithm operates similarly to Janner et al. (2019) and trains a policy from real and model-generated rollouts to maximize the extrinsic and intrinsic rewards. For the policy training, we use the soft actor-critic (SAC) algorithm (Haarnoja et al., 2018), and for the intrinsic reward coefficient, $\lambda_n$, we use the auto-tuning approach from Sukhija et al. (2024a). We train an ensemble of dynamics models and use their disagreement to quantify the epistemic uncertainty. As baselines, we consider (*i*) MBPO-MEAN, which maximizes only the extrinsic reward, i.e., $\lambda_n = 0$, and (*ii*) MBPO-PETS, which is based on the PETS algorithm (Chua et al., 2018) maximizing the extrinsic rewards in expectation over the ensemble dynamics (see Equation (5)). We report the results on the left side of Figure 3. We conclude that across all tasks, MBPO-OPTIMISTIC performs the best. Particularly, in sparse reward tasks such as the Mountaincar and CartPole, MBPO-OPTIMISTIC successfully solves the task whereas the greedy baselines fail. MBPO-OPTIMISTIC also successfully scales to high dimensional problems such as the Quadruped and Humanoid environments. We provide additional experiments with MBPO-OPTIMISTIC in Appendix C, where we evaluate it on more environments and compare it with pure off-policy algorithms SAC and MaxInfoRL (Sukhija et al., 2024a).

**Visual control experiments**  We investigate the scalability of SOMBRL to challenging and high-dimensional problems by evaluating it on visual control tasks. We combine SOMBRL with DREAMER (Hafner et al., 2023), an MBRL algorithm for visual control problems, and call the resulting algorithm DREAMER-OPTIMISTIC. We use the same approach as Sekar et al. (2020) for quantifying the epistemic uncertainty and for selecting the intrinsic reward coefficient, $\lambda_n$, we use the auto-tuning approach from Sukhija et al. (2024a). We report the results on the right side of Figure 3. Overall, DREAMER-OPTIMISTIC performs on-par with DREAMER on most tasks and outperforms it on the Finger-spin task from DMC and the Venture task from the Atari benchmark. Particularly, for Venture, a sparse reward task, Dreamer fails to achieve any reward. Curi et al. (2020) study the sensitivity of greedy exploration algorithms w.r.t. the action penalties in the reward. Inspired by their experiments, we modify the reward for the CartPole and Finger spin environments by adding an action cost, $r_{\text{action}}(\boldsymbol{a}) = -K \|\boldsymbol{a}\|_2$, where $K$ controls the penalty for large actions. Curi et al. (2020) show that even for small action costs, greedy exploration methods fail, converging to the sub-optimal solution of applying small actions. We observe a similar outcome in Figure 4 (left side), where DREAMER fails to solve the tasks for both the Finger spin and CartPole environments. On the other hand, DREAMER-OPTIMISTIC achieves much higher returns due to its optimistic exploration.

We provide additional experiments with DREAMER-OPTIMISTIC, including more environments and proprioceptive tasks in Appendix C.

**Hardware experiments**  Rothfuss et al. (2024) propose a novel approach for training deep Bayesian models that incorporates low-fidelity physical priors. Their approach significantly improves sample efficiency, which they illustrate in their hardware experiments on an RC car. Inspired by their experimental setup, we conduct a similar experiment on a highly dynamic RC car. The task is to perform a complex parking maneuver with drifting as depicted in Figure 1. We use the same experimental setup as Rothfuss et al. (2024).

First, we evaluate our algorithm SIMFSVGD-OPTIMISTIC, a combination of SOMBRL and SIMFSVGD, in simulation. The simulation is based on a realistic race car simulation from Kabzan et al. (2020). For the simulation experiments, we ablate different choices for the reward parameters, starting with the dense reward configuration from Rothfuss et al. (2024) and adapt parameters to obtain sparser rewards (see Appendix D for more detail). We report the results in the top row of Figure 4. For the dense reward setting, SIMFSVGD and SIMFSVGD-OPTIMISTIC perform similarly, but for sparser rewards, SIMFSVGD-OPTIMISTIC outperforms SIMFSVGD. In particular,

---

[4]including the humanoid from DMC: $d_{\boldsymbol{x}} = 67$, $d_{\boldsymbol{u}} = 21$

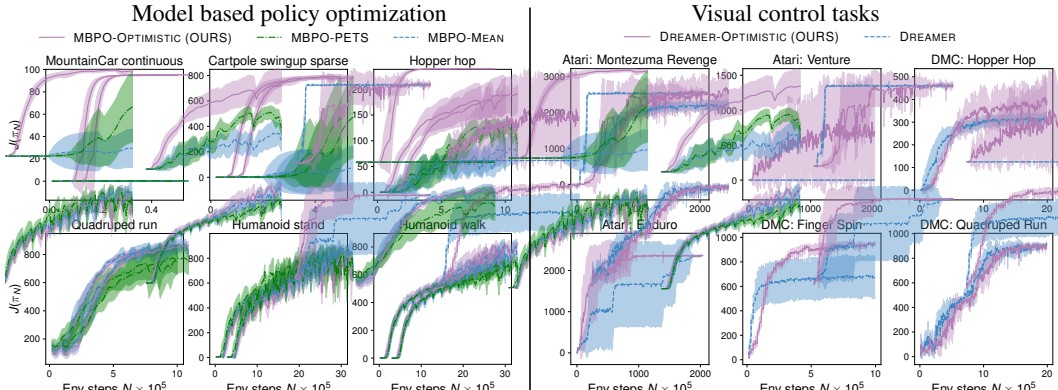

Figure 3: *Left:* Learning curves for the state-based tasks from DMC using MBPO as the base algorithm. Across all experiments, MBPO-OPTIMISTIC obtains the best performance compared to its greedy variants. MBPO-OPTIMISTIC also scales to high-dimensional tasks, specifically the humanoid environments from DMC. *Right:* Learning curves for the visual control tasks from DMC and Atari using DREAMER as the base algorithm. DREAMER-OPTIMISTIC either performs on-par or better than DREAMER in all our experiments. Particularly, in the Venture task from the Atari benchmark, where DREAMER fails to obtain any rewards.

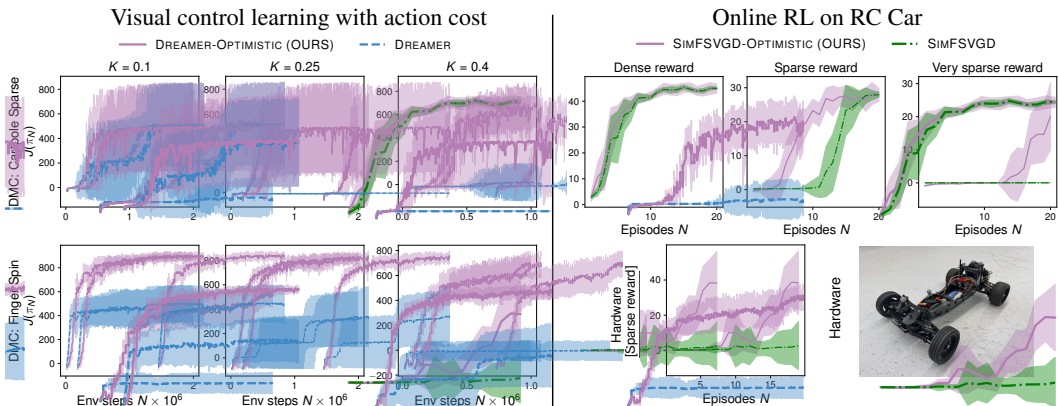

Figure 4: *Left:* Learning curves with action costs, where we compare DREAMER with DREAMER-OPTIMISTIC. DREAMER fails to explore sufficiently with action costs, whereas DREAMER-OPTIMISTIC is able to explore and obtain much higher performance. *Right:* Learning curves for our experiments with SIMFSVGD. *Top row*: We change the parameters of the reward function from Rothfuss et al. (2024), and make it sparse, starting from their dense reward. We observe that, as the reward gets sparser, SIMFSVGD drops in performance and SIMFSVGD-OPTIMISTIC outperforms it. *Bottom row*: We run the sparse reward configuration on hardware (depicted on the right side at the bottom), where we obtain similar results. As opposed to SIMFSVGD-OPTIMISTIC, SIMFSVGD fails to solve the task.

for the setting with very sparse rewards, SIMFSVGD completely fails to solve the task. We conduct our hardware experiments using the sparse reward configuration, and report the learning curve in the bottom row of Figure 4. In line with the simulation experiments, we also observe similar behavior on hardware. SIMFSVGD-OPTIMISTIC learns to solve the task, whereas SIMFSVGD completely fails. In fact, out of 5 attempts, SIMFSVGD worked only once, and otherwise converged to a local optimum of not moving from the starting position[5].

## 7 Conclusion

In this work, we propose SOMBRL, which maximizes a weighted sum of the extrinsic reward and the agent's epistemic uncertainty. We show that SOMBRL effectively performs optimistic exploration and provide regret bounds for it in a variety of settings, in particular for continuous state-action spaces

---

[5]Video is available at https://sukhijab.github.io/projects/sombrl/

and many common classes of RL problems, namely, finite-horizon, infinite-horizon, episodic, and non-episodic RL. To the best of our knowledge, we are the first to provide these theoretical guarantees, yielding a more flexible, scalable, and principled algorithm for exploration. We illustrate the strengths of SOMBRL in our experiments, where we combine SOMBRL with different model-based RL algorithms, evaluate it on tasks of varying dimensionality, including visual control problems, and also illustrate the benefits of optimistic exploration on hardware. In all cases, SOMBRL achieves the best performance, being significantly more scalable and computationally cheaper than prior optimistic exploration methods and stronger at exploration than SOTA deep RL baselines.

A limitation of SOMBRL is that it requires training a probabilistic model, e.g., deep ensembles, for quantifying epistemic uncertainty. We report the computational cost of our method in Appendix D.5 and show that the computational overhead is small compared to the training cost for the actor and critic.

**Acknowledgments**

We thank Scott Sussex for the insightful discussion and feedback on this work. This project has received funding from the Swiss National Science Foundation under NCCR Automation, grant agreement 51NF40 180545, the Microsoft Swiss Joint Research Center, and the SNSF Postdoc Mobility Fellowship 211086. Bhavya Sukhija was gratefully supported by ELSA (European Lighthouse on Secure and Safe AI) funded by the European Union under grant agreement No. 101070617.

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

# A Related Work

**Deep model-based RL**    Model-based RL algorithms offer a sample-efficient solution for learning directly in the real world Hansen et al. (2022); Wu et al. (2023); Rothfuss et al. (2024). Most widely applied algorithms (Chua et al., 2018; Janner et al., 2019; Hafner et al., 2023; Hansen et al., 2023) commonly rely on naive exploration techniques such as Boltzmann exploration and differ primarily in the type of dynamics modeling and policy planners. Cesa-Bianchi et al. (2017) show that Boltzmann exploration is suboptimal even in the simplified setting of stochastic bandits. SOMBRL is agnostic to the choice of modeling and planners, as we demonstrate in Section 5 and 6. Moreover, we focus on the problem of exploration for MBRL and propose a principled exploration approach. We derive regret bounds for our approach, showing that it is theoretically grounded. Furthermore, we illustrate the benefits of principled exploration in our hardware experiment, where the naive exploration baseline fails to obtain any meaningful exploration. To the best of our knowledge, we are the first to propose a simple, flexible, scalable, and theoretically grounded approach for principled exploration and show its benefits directly in the real world.

**Theoretical results for Model-based RL**    There are numerous works that study MBRL for linear dynamical systems theoretically (Abbasi-Yadkori & Szepesvári, 2011; Cohen et al., 2019; Simchowitz & Foster, 2020; Dean et al., 2020; Faradonbeh et al., 2020; Abeille & Lazaric, 2020; Treven et al., 2021), focusing primarily on the challenges of nonepisodic learning. In the nonlinear case, Kakade et al. (2020); Curi et al. (2020); Mania et al. (2020); Wagenmaker et al. (2023); Treven et al. (2024) analyze the finite-horizon episodic setting and provide regret bounds that are sublinear for many RKHS. Recently, Sukhija et al. (2024b) extended these results to the nonepisodic setting. Crucially, most of these algorithms are based on the principle of optimism in the face of uncertainty and require solving the problem in Equation (7). As highlighted in Section 1 and 3, solving these problems is often intractable or computationally expensive. Therefore, naive exploration techniques, such as Boltzmann exploration are more widely used. SOMBRL addresses this drawback and proposes an alternative optimistic exploration method, which is much simpler and more scalable. Furthermore, it enjoys the same asymptotic guarantees as these methods and hence is also theoretically grounded.

**Intrinsic exploration in RL**    Intrinsic rewards are often used as a surrogate objective for principled exploration in challenging tasks (see Aubret et al., 2023, for a comprehensive survey). Common choices of intrinsic rewards are model prediction error or "Curiosity" (Schmidhuber, 1991; Pathak et al., 2017; Burda et al., 2018), novelty of transitions/state-visitation counts (Stadie et al., 2015; Bellemare et al., 2016), diversity of skills/goals (Eysenbach et al., 2018; Sharma et al., 2019; Nair et al., 2018; Pong et al., 2019), empowerment (Klyubin et al., 2005; Salge et al., 2014), and information gain of the dynamics (Sekar et al., 2020; Mendonca et al., 2021; Sukhija et al., 2024c). However, these rewards are mostly used for pure exploration and rarely considered in combination with the extrinsic reward. We show that combining the model epistemic uncertainty, an intrinsic reward, with the extrinsic one, effectively performs optimistic exploration, thus, providing a theoretical grounding for our approach. There are a few works from bandits (Auer, 2002; Srinivas et al., 2012), data-driven control (Åström & Wittenmark, 1971; Chiuso et al., 2023; Grimaldi et al., 2024), and RL (Abeille & Lazaric, 2020; Sukhija et al., 2024a) that have also proposed maximizing extrinsic rewards jointly with epistemic uncertainty. The data-driven control community refers to this as the separation principle between model identification and control design (Åström & Wittenmark, 1971; Chiuso et al., 2023; Grimaldi et al., 2024). In RL, Abeille & Lazaric (2020) show duality between Equation (7) and Equation (8) for linear systems. For nonlinear systems and deep RL, Sukhija et al. (2024a) empirically study combining extrinsic and intrinsic rewards. However, compared to these works, we demonstrate SOMBRL's scalability to high-dimensional settings such as visual control tasks, and additionally, ground this approach theoretically, providing regret bounds for nonlinear systems and common RL settings.

# B  Proofs

## B.1  Definition of Well-calibrated Model

We define a well-calibrated statistical model of $\boldsymbol{f}^*$, which captures both the mean prediction $\boldsymbol{\mu}_n$ and the uncertainty $\boldsymbol{\sigma}_n$ of our learned model.

**Definition B.1** (Well-calibrated statistical model of $\boldsymbol{f}^*$, Rothfuss et al. (2023)). *Let $\mathcal{Z} \stackrel{\text{def}}{=} \mathcal{X} \times \mathcal{U}$. A sequence of sets $\{\mathcal{M}_n(\delta)\}_{n \geq 0}$, where*

$$\mathcal{M}_n(\delta) \stackrel{\text{def}}{=} \left\{ \boldsymbol{f} : \mathcal{Z} \to \mathbb{R}^{d_{\boldsymbol{x}}} \mid \forall \boldsymbol{z} \in \mathcal{Z}, \forall j \in \{1, \ldots, d_{\boldsymbol{x}}\} : |\mu_{n,j}(\boldsymbol{z}) - f_j(\boldsymbol{z})| \leq \beta_n(\delta)\sigma_{n,j}(\boldsymbol{z}) \right\},$$

*is an all-time well-calibrated statistical model of the function $\boldsymbol{f}^*$, if, with probability at least $1 - \delta$, we have $\boldsymbol{f}^* \in \bigcap_{n \geq 0} \mathcal{M}_n(\delta)$. Here, $f_j$, $\mu_{n,j}$ and $\sigma_{n,j}$ denote the $j$-th element in the vector-valued functions $\boldsymbol{f}$, $\boldsymbol{\mu}_n$ and $\boldsymbol{\sigma}_n$ respectively, and $\beta_n(\delta) \in \mathbb{R}_{\geq 0}$ is a scalar function that depends on the confidence level $\delta \in (0, 1]$ and which is monotonically increasing in $n$.*

In the following, we show that Gaussian processes are well-callibrated models if Assumption 5.2 holds.

**Lemma B.2** (Well calibrated confidence intervals for RKHS, Rothfuss et al. (2023)). *Let $\boldsymbol{f}^* \in \mathcal{H}_{k,B}^{d_{\boldsymbol{x}}}$. Suppose $\boldsymbol{\mu}_n$ and $\boldsymbol{\sigma}_n$ are the posterior mean and variance of a GP with kernel $k$, Equation (9). There exists $\beta_n(\delta)$, for which the tuple $(\boldsymbol{\mu}_n, \boldsymbol{\sigma}_n, \beta_n(\delta))$ is a well-calibrated statistical model of $\boldsymbol{f}^*$.*

In summary, in the RKHS setting, a GP is a well-calibrated model.

## B.2  Analysis for the finite horizon case

**Lemma B.3.** *Let Assumption 5.1 and Assumption 5.2 hold. Consider the following definitions*

$$J(\boldsymbol{\pi}, \boldsymbol{f}^*) = \mathbb{E}_{\boldsymbol{f}^*}\left[\sum_{t=0}^{T-1} r(\boldsymbol{x}_t, \boldsymbol{\pi}(\boldsymbol{x}_t))\right], \ s.t., \ \boldsymbol{x}_{t+1} = \boldsymbol{f}^*(\boldsymbol{x}_t, \boldsymbol{\pi}(\boldsymbol{x}_t)) + \boldsymbol{w}_t, \quad \boldsymbol{x}_0 = \boldsymbol{x}(0).$$

$$J(\boldsymbol{\pi}, \boldsymbol{\mu}_n) = \mathbb{E}_{\boldsymbol{\mu}_n}\left[\sum_{t=0}^{T-1} r(\boldsymbol{x}_t', \boldsymbol{\pi}(\boldsymbol{x}_t'))\right], \ s.t., \ \boldsymbol{x}_{t+1}' = \boldsymbol{\mu}_n(\boldsymbol{x}_t', \boldsymbol{\pi}(\boldsymbol{x}_t')) + \boldsymbol{w}_t, \quad \boldsymbol{x}_0' = \boldsymbol{x}(0).$$

$$\Sigma_n(\boldsymbol{\pi}, \boldsymbol{f}^*) = \mathbb{E}_{\boldsymbol{f}^*}\left[\sum_{t=0}^{T-1} \|\boldsymbol{\sigma}_n(\boldsymbol{x}_t, \boldsymbol{\pi}(\boldsymbol{x}_t))\|\right], \ s.t., \ \boldsymbol{x}_{t+1} = \boldsymbol{f}^*(\boldsymbol{x}_t, \boldsymbol{\pi}(\boldsymbol{x}_t)) + \boldsymbol{w}_t, \quad \boldsymbol{x}_0 = \boldsymbol{x}(0).$$

$$\Sigma_n(\boldsymbol{\pi}, \boldsymbol{\mu}_n) = \mathbb{E}_{\boldsymbol{\mu}_n}\left[\sum_{t=0}^{T-1} \|\boldsymbol{\sigma}_n(\boldsymbol{x}_t', \boldsymbol{\pi}(\boldsymbol{x}_t'))\|\right]. \ s.t., \ \boldsymbol{x}_{t+1}' = \boldsymbol{\mu}_n(\boldsymbol{x}_t', \boldsymbol{\pi}(\boldsymbol{x}_t')) + \boldsymbol{w}_t, \quad \boldsymbol{x}_0' = \boldsymbol{x}(0).$$

$$\lambda_n = C_{\max} T \frac{(1 + \sqrt{d_x})\beta_{n-1}(\delta)}{\sigma},$$

*where $C_{\max} = \max\{R_{\max}, \sigma_{\max}\}$. Then we have for all $n \geq 0$, $\boldsymbol{\pi} \in \Pi$ with probability at least $1 - \delta$*

$$|J(\boldsymbol{\pi}, \boldsymbol{f}^*) - J(\boldsymbol{\pi}, \boldsymbol{\mu}_n)| \leq \lambda_n \Sigma_n(\boldsymbol{\pi}, \boldsymbol{\mu}_n)$$
$$|J(\boldsymbol{\pi}, \boldsymbol{f}^*) - J(\boldsymbol{\pi}, \boldsymbol{\mu}_n)| \leq \lambda_n \Sigma_n(\boldsymbol{\pi}, \boldsymbol{f}^*)$$

*Proof.* We give the proof for $|J(\boldsymbol{\pi}, \boldsymbol{f}^*) - J(\boldsymbol{\pi}, \boldsymbol{\mu}_n)| \leq \lambda_n(L_r, \boldsymbol{\mu}_n)\Sigma_n(\boldsymbol{\pi}, \boldsymbol{\mu}_n)$. The same argument holds for the second inequality. Let $J_{t+1}(\boldsymbol{\pi}, \boldsymbol{f}^*, \boldsymbol{x})$ denote the cost-to-go from state $\boldsymbol{x}$, step $t + 1$ onwards under the dynamics $\boldsymbol{f}^*$. Following the Policy difference Lemma from (Kakade & Langford, 2002) and Sukhija et al. (2024c, Corollary 2.)

$$J(\boldsymbol{\pi}, \boldsymbol{\mu}_n) - J(\boldsymbol{\pi}, \boldsymbol{f}^*) = \mathbb{E}_{\boldsymbol{\mu}_n}\left[\sum_{t=0}^{T-1} J_{t+1}(\boldsymbol{\pi}, \boldsymbol{f}^*, \boldsymbol{x}_{t+1}') - J_{t+1}(\boldsymbol{\pi}, \boldsymbol{f}^*, \hat{\boldsymbol{x}}_{t+1})\right],$$

with $\hat{\boldsymbol{x}}_{t+1} = \boldsymbol{f}^*(\boldsymbol{x}_t', \boldsymbol{\pi}(\boldsymbol{x}_t')) + \boldsymbol{w}_t$, and $\boldsymbol{x}_{t+1}' = \boldsymbol{\mu}_n(\boldsymbol{x}_t', \boldsymbol{\pi}(\boldsymbol{x}_t')) + \boldsymbol{w}_t$.

Therefore,

$$|J(\boldsymbol{\pi}, \boldsymbol{\mu}_n) - J(\boldsymbol{\pi}, \boldsymbol{f}^*)| = \left|\mathbb{E}\left[\sum_{t=0}^{T-1} J_{t+1}(\boldsymbol{\pi}, \boldsymbol{f}^*, \boldsymbol{x}_{t+1}') - J_{t+1}(\boldsymbol{\pi}, \boldsymbol{f}^*, \hat{\boldsymbol{x}}_{t+1})\right]\right|$$

$$\leq \sum_{t=0}^{T-1} \mathbb{E}\left[\left|\mathbb{E}_{\boldsymbol{w}_t}\left[J_{t+1}(\boldsymbol{\pi}, \boldsymbol{f}^*, \boldsymbol{x}'_{t+1}) - J_{t+1}(\boldsymbol{\pi}, \boldsymbol{f}^*, \hat{\boldsymbol{x}}_{t+1})\right]\right|\right]$$

Next, we bound the last term using the derivation from Kakade et al. (2020). Let $C(\boldsymbol{x}) = J_{t+1}^2(\boldsymbol{\pi}, \boldsymbol{f}^*, \boldsymbol{x})$.

$$\left|\mathbb{E}_{\boldsymbol{w}_t}\left[J_{t+1}(\boldsymbol{\pi}, \boldsymbol{f}^*, \boldsymbol{x}'_{t+1}) - J_{t+1}(\boldsymbol{\pi}, \boldsymbol{f}^*, \hat{\boldsymbol{x}}_{t+1})\right]\right|$$

$$\leq \sqrt{\max\left\{\mathbb{E}_{\boldsymbol{w}_t}[C(\boldsymbol{x}'_{t+1})], \mathbb{E}_{\boldsymbol{w}_t}[C(\hat{\boldsymbol{x}}_{t+1})]\right\}}$$

$$\times \min\left\{\frac{\|\boldsymbol{f}^*(\boldsymbol{x}'_t, \boldsymbol{\pi}(\boldsymbol{x}'_t)) - \boldsymbol{\mu}_n(\boldsymbol{x}'_t, \boldsymbol{\pi}(\boldsymbol{x}'_t))\|}{\sigma}, 1\right\} \qquad \text{(Kakade et al., 2020, Lemma C.2.)}$$

$$\leq R_{\max}T \frac{(1 + \sqrt{d_x})\beta_{n-1}(\delta)}{\sigma}\|\boldsymbol{\sigma}_{n-1}(\boldsymbol{x}'_t, \boldsymbol{\pi}(\boldsymbol{x}'_t))\|$$

Therefore, we have

$$|J(\boldsymbol{\pi}, \boldsymbol{\mu}_n) - J(\boldsymbol{\pi}, \boldsymbol{f}^*)|$$

$$\leq \sum_{t=0}^{T-1} \mathbb{E}\left[\left|\mathbb{E}_{\boldsymbol{w}_t}\left[J_{t+1}(\boldsymbol{\pi}, \boldsymbol{f}^*, \boldsymbol{x}'_{t+1}) - J_{t+1}(\boldsymbol{\pi}, \boldsymbol{f}^*, \hat{\boldsymbol{x}}_{t+1})\right]\right|\right]$$

$$\leq \lambda_n \sum_{t=0}^{T-1} \mathbb{E}\left[\|\boldsymbol{\sigma}_{n-1}(\boldsymbol{x}'_t, \boldsymbol{\pi}(\boldsymbol{x}'_t))\|\right].$$

$\square$

Note that Lemma 5.3 follows directly from Lemma B.3.

**Lemma B.4.** *Let Assumption 5.1 and Assumption 5.2 hold and consider the simple regret at episode $n$, $r_n = J(\boldsymbol{\pi}^*, \boldsymbol{f}^*) - J(\boldsymbol{\pi}_n, \boldsymbol{f}^*)$. The following holds for all $n > 0$ with probability at least $1 - \delta$*

$$r_n \leq (2\lambda_n + \lambda_n^2)\Sigma_n(\boldsymbol{\pi}_n, \boldsymbol{f}^*)$$

*Proof.*

$$\begin{aligned}
r_n &= J(\boldsymbol{\pi}^*, \boldsymbol{f}^*) - J(\boldsymbol{\pi}_n, \boldsymbol{f}^*) \\
&\leq J(\boldsymbol{\pi}^*, \boldsymbol{\mu}_n) + \lambda_n \Sigma_n(\boldsymbol{\pi}^*, \boldsymbol{\mu}_n) - J(\boldsymbol{\pi}_n, \boldsymbol{f}^*) && \text{(Lemma B.3)} \\
&\leq J(\boldsymbol{\pi}_n, \boldsymbol{\mu}_n) + \lambda_n \Sigma_n(\boldsymbol{\pi}_n, \boldsymbol{\mu}_n) - J(\boldsymbol{\pi}_n, \boldsymbol{f}^*) && \text{(Equation (8))} \\
&= J(\boldsymbol{\pi}_n, \boldsymbol{\mu}_n) - J(\boldsymbol{\pi}_n, \boldsymbol{f}^*) + \lambda_n \Sigma_n(\boldsymbol{\pi}_n, \boldsymbol{\mu}_n) \\
&\leq \lambda_n \Sigma_n(\boldsymbol{\pi}_n, \boldsymbol{f}^*) + \lambda_n \Sigma_n(\boldsymbol{\pi}_n, \boldsymbol{\mu}_n) && \text{(Lemma B.3)} \\
&= 2\lambda_n \Sigma_n(\boldsymbol{\pi}_n, \boldsymbol{f}^*) + \lambda_n (\Sigma_n(\boldsymbol{\pi}_n, \boldsymbol{\mu}_n) - \Sigma_n(\boldsymbol{\pi}_n, \boldsymbol{f}^*)) \\
&\leq (\lambda_n^2 + 2\lambda_n)\Sigma_n(\boldsymbol{\pi}_n, \boldsymbol{f}^*).
\end{aligned}$$

Here in the last inequality, we used the fact that $\|\boldsymbol{\sigma}(\cdot, \cdot)\|$ is bounded and positive, therefore, we can treat it similar to the reward (it is in fact an intrinsic reward) and use Lemma B.3. $\square$

*Proof of Theorem 5.4.*

$$\begin{aligned}
R_N &= \sum_{n=1}^{N} r_n \\
&\leq \sum_{n=1}^{N} (\lambda_n^2 + 2\lambda_n)\Sigma_n(\boldsymbol{\pi}_n, \boldsymbol{f}^*) \\
&\leq (\lambda_N^2 + \lambda_N) \sum_{n=1}^{N} \Sigma_n(\boldsymbol{\pi}_n, \boldsymbol{f}^*)
\end{aligned}$$

$$= (\lambda_N^2 + 2\lambda_N) \sum_{n=1}^{N} \mathbb{E}_{\boldsymbol{f}^*} \left[ \sum_{t=0}^{T-1} \|\boldsymbol{\sigma}_n(\boldsymbol{x}_t, \boldsymbol{\pi}(\boldsymbol{x}_t))\| \right]$$

$$\leq (\lambda_N^2 + 2\lambda_N) \sqrt{NT} \sum_{n=1}^{N} \mathbb{E}_{\boldsymbol{f}^*} \left[ \sum_{t=0}^{T-1} \|\boldsymbol{\sigma}_n^2(\boldsymbol{x}_t, \boldsymbol{\pi}(\boldsymbol{x}_t))\| \right]$$

$$\leq C(\lambda_N^2 + 2\lambda_N) T \sqrt{N\Gamma_{NT}} \qquad \text{(Curi et al. (2020), Lemma 17))}$$

Finally, note that from Lemma B.3 we have $\lambda_N \propto T\beta_n$ and $\beta_n \propto \sqrt{\Gamma_n}$ (Chowdhury & Gopalan, 2017). Therefore, $R_N \leq \mathcal{O}(T^3 \Gamma_N^{3/2} \sqrt{N})$ $\qquad\qquad\square$

Table 2: Maximum information gain bounds for common choice of kernels.

| Kernel | $k(\boldsymbol{x}, \boldsymbol{x}')$ | $\Gamma_N$ |
|---|---|---|
| Linear | $\boldsymbol{x}^\top \boldsymbol{x}'$ | $\mathcal{O}\left(d \log(N)\right)$ |
| RBF | $e^{-\frac{\|\boldsymbol{x}-\boldsymbol{x}'\|^2}{2l^2}}$ | $\mathcal{O}\left(\log^{d+1}(N)\right)$ |
| Matèrn | $\frac{1}{\Gamma(\nu)2^{\nu-1}} \left( \frac{\sqrt{2\nu}\|\boldsymbol{x}-\boldsymbol{x}'\|}{l} \right)^\nu B_\nu \left( \frac{\sqrt{2\nu}\|\boldsymbol{x}-\boldsymbol{x}'\|}{l} \right)$ | $\mathcal{O}\left(N^{\frac{d}{2\nu+d}} \log^{\frac{2\nu}{2\nu+d}}(N)\right)$ |

In Table 2 we list rates of $\Gamma_N$ for the most common choice of kernels.

### B.3 Analysis for the discounted infinite horizon case

For the infinite horizon case, we first study the posterior variance $\boldsymbol{\sigma}_n$ in the feature space. Moreover, let $\boldsymbol{z} = (\boldsymbol{x}, \boldsymbol{u})$ and $\mathcal{Z} = \mathcal{X} \times \mathcal{U}$.

For the ease of notation we denote $\boldsymbol{z}_{k,n} = (\boldsymbol{x}_k^n, \boldsymbol{\pi}_n(\boldsymbol{x}_k^n))$. For $\boldsymbol{z}$ we define the kernel embedding $k_{\boldsymbol{z}} = k(\boldsymbol{z}, \cdot)$. The covariance matrix $\boldsymbol{V}_t : \mathcal{H} \to \mathcal{H}$ in the feature form is:

$$\boldsymbol{V}_t = \boldsymbol{I} + \frac{1}{\sigma^2} \sum_{i=1}^{t} k_{\boldsymbol{z}_i} k_{\boldsymbol{z}_i}^\top. \tag{10}$$

Note that we have $\boldsymbol{x}_{t+1} = \langle k_{\boldsymbol{z}_t}, \boldsymbol{f}^* \rangle_{\mathcal{H}} + \boldsymbol{w}_t$. With the design matrix $\boldsymbol{M}_t : \mathcal{H} \to \mathbb{R}^t$

$$\boldsymbol{M}_t = (k_{\boldsymbol{z}_1} \quad k_{\boldsymbol{z}_2} \quad \cdots \quad k_{\boldsymbol{z}_t}) \tag{11}$$

we have $\boldsymbol{V}_t = \boldsymbol{I} + \frac{1}{\sigma^2} \boldsymbol{M}_t \boldsymbol{M}_t^\top$ and since $\boldsymbol{K}_t = \boldsymbol{M}_t^\top \boldsymbol{M}_t$ we have

$$\det(\boldsymbol{V}_t) = \det\left(\boldsymbol{I} + \frac{1}{\sigma^2} \boldsymbol{K}_t\right) \tag{12}$$

**Corollary B.5** (Lower bound on the posterior log determinant).

$$\log\left(|\boldsymbol{V}_n|\right) \geq \log\left(|\boldsymbol{V}_{n-1}|\right) + \log\left(1 + \sigma^{-2} \sum_{k=1}^{\widehat{T}_n} \|\boldsymbol{\sigma}_{n-1}(\boldsymbol{z}_{k,n})\|^2\right) \tag{13}$$

*In particular, we have*

$$\log\left(\frac{|\boldsymbol{V}_N|}{|\boldsymbol{V}_0|}\right) \geq \sum_{n=1}^{N} \log\left(1 + \sigma^{-2} \sum_{k=1}^{\widehat{T}_n} \|\boldsymbol{\sigma}_{n-1}(\boldsymbol{z}_{k,n})\|^2\right) \tag{14}$$

*Proof.*

$$\log\left(|\boldsymbol{V}_n|\right) = \log\left(|\boldsymbol{V}_{n-1}|\right) + \log\left(\left| \boldsymbol{I} + \sigma^{-2} \boldsymbol{V}_{n-1}^{-1/2} \sum_{k=1}^{\widehat{T}_n} \boldsymbol{k}_{\boldsymbol{z}_{k,n}} \boldsymbol{k}_{\boldsymbol{z}_{k,n}}^\top \boldsymbol{V}_{n-1}^{-1/2} \right| \right)$$

$$\geq \log\left(|\boldsymbol{V}_{n-1}|\right)$$

$$+ \log\left(1 + \mathrm{tr}\left(\sigma^{-2}\boldsymbol{V}_{n-1}^{-1/2}\sum_{k=1}^{\widehat{T}_n}\boldsymbol{k}_{\boldsymbol{z}_{k,n}}\boldsymbol{k}_{\boldsymbol{z}_{k,n}}^{\top}\boldsymbol{V}_{n-1}^{-1/2}\right)\right) \qquad \text{(see (*) below)}$$

$$= \log\left(|\boldsymbol{V}_{n-1}|\right) + \log\left(1 + \sigma^{-2}\sum_{k=1}^{\widehat{T}_n}\left\|\boldsymbol{k}_{\boldsymbol{z}_{k,n}}\right\|_{\boldsymbol{V}_{n-1}^{-1}}^{2}\right)$$

$$= \log\left(|\boldsymbol{V}_{n-1}|\right) + \log\left(1 + \sigma^{-2}\sum_{k=1}^{\widehat{T}_n}\left\|\boldsymbol{\sigma}_{n-1}(\boldsymbol{z}_{k,n})\right\|^{2}\right)$$

We prove (*) in the following, first let $\boldsymbol{m}_k = \sigma^{-1}\boldsymbol{V}_{n-1}^{-1/2}\boldsymbol{k}_{\boldsymbol{z}_{k,n}}$, then we have

$$\log\left(\left|\boldsymbol{I} + \sigma^{-2}\boldsymbol{V}_{n-1}^{-1/2}\sum_{k=1}^{\widehat{T}_n}\boldsymbol{k}_{\boldsymbol{z}_{k,n}}\boldsymbol{k}_{\boldsymbol{z}_{k,n}}^{\top}\boldsymbol{V}_{n-1}^{-1/2}\right|\right) = \log\left(\left|\boldsymbol{I} + \sum_{k=1}^{\widehat{T}_n}\boldsymbol{m}_k\boldsymbol{m}_k^{\top}\right|\right).$$

The matrix $\boldsymbol{M} = \sum_{k=1}^{\widehat{T}_n}\boldsymbol{m}_k\boldsymbol{m}_k^{\top}$ by definition is positive semi-definite. Moreover, $|\boldsymbol{I} + \boldsymbol{M}| = \prod_{i\geq 1}(1 + \alpha_i)$, where $\alpha_i \geq 0$ are the eigenvalues of $\boldsymbol{M}$. Furthermore, since $\alpha_i \geq 0$ and $\prod_{i\geq 1}(1 + \alpha_i) = 1 + \sum_{i\geq 1}\alpha_i + \cdots + \prod_{i\geq 1}\alpha_i$, we get $\prod_{i\geq 1}(1 + \alpha_i) \geq 1 + \sum_{i\geq 1}\alpha_i$. Finally, since $\sum_{i\geq 1}\alpha_i = \mathrm{tr}\,(\boldsymbol{M})$, we get $|\boldsymbol{I} + \boldsymbol{M}| \geq 1 + \mathrm{tr}\,(\boldsymbol{M})$. $\qquad\square$

**Corollary B.6** (Upper bound on the posterior log determinant)**.**

$$\log\left(|\boldsymbol{V}_n|\right) \leq \log\left(|\boldsymbol{V}_{n-1}|\right) + \sum_{k=1}^{\widehat{T}_n}\sum_{j=1}^{d_x}\log\left(1 + \sigma^{-2}\sigma_{n-1,j}^2(\boldsymbol{z}_{k,n})\right)$$

*Proof.*

$$\log\left(|\boldsymbol{V}_n|\right) = \log\left(|\boldsymbol{V}_{n-1}|\right) + \log\left(|\boldsymbol{I} + \boldsymbol{M}|\right)$$
$$\leq \log\left(|\boldsymbol{V}_{n-1}|\right) + \log\left(|\mathrm{diag}\,(\boldsymbol{I} + \boldsymbol{M})|\right) \qquad \text{(Hadamard's inequality for PSD matrices)}$$
$$= \log\left(|\boldsymbol{V}_{n-1}|\right) + \sum_{k=1}^{\widehat{T}_n}\sum_{j=1}^{d_x}\log\left(1 + \sigma^{-2}\sigma_{n-1,j}^2(\boldsymbol{z}_{k,n})\right)$$

$\qquad\square$

Corollary B.5 will be useful for the discounted horizon case, whereas Corollary B.6 will be applied for the nonepisodic setting.

Next, we show that SOMBRL also performs optimism in the discounted horizon case.
**Lemma B.7.** *Let Assumption 5.1, and Assumption 5.2 hold. Consider the following definitions*

$$J_\gamma(\boldsymbol{\pi}, \boldsymbol{f}^*) = \mathbb{E}\left[\sum_{t=0}^{\infty}\gamma^t r(\boldsymbol{x}_t, \boldsymbol{\pi}(\boldsymbol{x}_t))\right] \; s.t., \; \boldsymbol{x}_{t+1} = \boldsymbol{f}^*(\boldsymbol{x}_t, \boldsymbol{\pi}(\boldsymbol{x}_t)) + \boldsymbol{w}_t, \quad \boldsymbol{x}_0 = \boldsymbol{x}(0),$$

$$J_\gamma(\boldsymbol{\pi}, \boldsymbol{\mu}_n) = \mathbb{E}\left[\sum_{t=0}^{\infty}\gamma^t r(\boldsymbol{x}_t', \boldsymbol{\pi}(\boldsymbol{x}_t'))\right] \; s.t., \; \boldsymbol{x}_{t+1}' = \boldsymbol{\mu}_n(\boldsymbol{x}_t', \boldsymbol{\pi}(\boldsymbol{x}_t')) + \boldsymbol{w}_t, \quad \boldsymbol{x}_0' = \boldsymbol{x}(0),$$

$$\Sigma_n^\gamma(\boldsymbol{\pi}, \boldsymbol{f}^*) = \mathbb{E}\left[\sum_{t=0}^{\infty}\gamma^t \left\|\boldsymbol{\sigma}_n(\boldsymbol{x}_t, \boldsymbol{\pi}(\boldsymbol{x}_t))\right\|\right] \; s.t., \; \boldsymbol{x}_{t+1} = \boldsymbol{f}^*(\boldsymbol{x}_t, \boldsymbol{\pi}(\boldsymbol{x}_t)) + \boldsymbol{w}_t, \quad \boldsymbol{x}_0 = \boldsymbol{x}(0),$$

$$\Sigma_n^\gamma(\boldsymbol{\pi}, \boldsymbol{\mu}_n) = \mathbb{E}\left[\sum_{t=0}^{\infty}\gamma^t \left\|\boldsymbol{\sigma}_n(\boldsymbol{x}_t', \boldsymbol{\pi}(\boldsymbol{x}_t'))\right\|\right] \; s.t., \; \boldsymbol{x}_{t+1}' = \boldsymbol{\mu}_n(\boldsymbol{x}_t', \boldsymbol{\pi}(\boldsymbol{x}_t')) + \boldsymbol{w}_t, \quad \boldsymbol{x}_0' = \boldsymbol{x}(0),$$

$$\lambda_n = C_{\max}\frac{\gamma}{1 - \gamma}\frac{(1 + \sqrt{d_x})\beta_{n-1}(\delta)}{\sigma},$$

where $C_{\max} = \max\{R_{\max}, \sigma_{\max}\}$. *Then we have for all $n \geq 0$, $\boldsymbol{\pi} \in \Pi$ with probability at least $1 - \delta$*

$$|J_\gamma(\boldsymbol{\pi}, \boldsymbol{f}^*) - J_\gamma(\boldsymbol{\pi}, \boldsymbol{\mu}_n)| \leq \lambda_n \Sigma_n^\gamma(\boldsymbol{\pi}, \boldsymbol{\mu}_n)$$
$$|J_\gamma(\boldsymbol{\pi}, \boldsymbol{f}^*) - J_\gamma(\boldsymbol{\pi}, \boldsymbol{\mu}_n)| \leq \lambda_n \Sigma_n^\gamma(\boldsymbol{\pi}, \boldsymbol{f}^*)$$

*Proof.* We give the proof for $|J_\gamma(\boldsymbol{\pi}, \boldsymbol{f}^*) - J_\gamma(\boldsymbol{\pi}, \boldsymbol{\mu}_n)| \leq \lambda_n(L_r, \boldsymbol{\mu}_n)\Sigma_n^\gamma(\boldsymbol{\pi}, \boldsymbol{\mu}_n)$. The same argument holds for the second inequality. We can extend the result from Sukhija et al. (2024a, Corollary 2.,) to the discounted case and get

$$J_\gamma(\boldsymbol{\pi}, \boldsymbol{\mu}_n) - J_\gamma(\boldsymbol{\pi}, \boldsymbol{f}^*) = \mathbb{E}_{\boldsymbol{\mu}_n}\left[\sum_{t=0}^\infty \gamma^{t+1}(J_\gamma(\boldsymbol{\pi}, \boldsymbol{f}^*, \boldsymbol{x}'_{t+1}) - J_\gamma(\boldsymbol{\pi}, \boldsymbol{f}^*, \hat{\boldsymbol{x}}_{t+1}))\right],$$

with $\hat{\boldsymbol{x}}_{t+1} = \boldsymbol{f}^*(\boldsymbol{s}'_t, \boldsymbol{\pi}(\boldsymbol{x}'_t)) + \boldsymbol{w}_t$, and $\boldsymbol{x}'_{t+1} = \boldsymbol{\mu}_n(\boldsymbol{s}'_t, \boldsymbol{\pi}(\boldsymbol{x}'_t)) + \boldsymbol{w}_t$.

Let $\beta_n \frac{1 + \sqrt{d_x}}{\sigma} C(\boldsymbol{x}) = J_\gamma^2(\boldsymbol{\pi}, \boldsymbol{f}^*, \boldsymbol{x})$. Note that $C(\boldsymbol{x}) \leq \lambda_n$ for all $\boldsymbol{c} \in \mathcal{X}$. Therefore, we have

$$|J_\gamma(\boldsymbol{\pi}, \boldsymbol{\mu}_n) - J_\gamma(\boldsymbol{\pi}, \boldsymbol{f}^*)|$$

$$= \left|\mathbb{E}\left[\sum_{t=0}^\infty \gamma^{t+1}(J_\gamma(\boldsymbol{\pi}, \boldsymbol{f}^*, \boldsymbol{x}'_{t+1}) - J_\gamma(\boldsymbol{\pi}, \boldsymbol{f}^*, \hat{\boldsymbol{x}}_{t+1}))\right]\right|$$

$$\leq \sum_{t=0}^\infty \gamma^{t+1}\mathbb{E}\left[|\mathbb{E}_{\boldsymbol{w}_t}\left[J_\gamma(\boldsymbol{\pi}, \boldsymbol{f}^*, \boldsymbol{x}'_{t+1}) - J_\gamma(\boldsymbol{\pi}, \boldsymbol{f}^*, \hat{\boldsymbol{x}}_{t+1})\right]|\right]$$

$$\leq \sum_{t=0}^\infty \gamma\mathbb{E}\left[\sqrt{\max\left\{\mathbb{E}_{\boldsymbol{w}_t}[C(\boldsymbol{x}'_{t+1})], \mathbb{E}_{\boldsymbol{w}_t}[C(\hat{\boldsymbol{x}}_{t+1})]\right\}}\right.$$

$$\left. \times \gamma^t \min\left\{\frac{\|\boldsymbol{f}^*(\boldsymbol{x}'_t, \boldsymbol{\pi}(\boldsymbol{x}'_t)) - \boldsymbol{\mu}_n(\boldsymbol{x}'_t, \boldsymbol{\pi}(\boldsymbol{x}'_t))\|}{\sigma}, 1\right\}\right] \qquad \text{(Kakade et al., 2020, Lemma C.2.)}$$

$$\leq \lambda_n \sum_{t=0}^\infty \mathbb{E}\left[\gamma^t \|\boldsymbol{\sigma}_{n-1}(\boldsymbol{x}'_t, \boldsymbol{\pi}(\boldsymbol{x}'_t))\|\right] \qquad \text{((Sukhija et al., 2024c, Corollary 3))}$$

$\square$

*Proof of Theorem 5.5.* We start with bounding

$$\sum_{n=1}^N \sum_{t=0}^\infty \mathbb{E}_{\boldsymbol{w}_{1:t-1}}\left[\gamma^t \|\boldsymbol{\sigma}_{n-1}(\boldsymbol{x}'_t, \boldsymbol{\pi}(\boldsymbol{x}'_t))\|^2\right] \qquad (15)$$

To achieve this, we use a sampling strategy where we increase the horizon of rollouts with each episode $n$. In the discounted setting, this allows us to collect data at the tails of our rollouts, i.e., make observations with longer rollouts and thus approximate the true discounted value function asymptotically. Moreover, we set $T(n) = -\frac{\log(n)}{\log(\gamma)}$ (note that $\gamma < 1$ and therefore $T(n)$ is positive).

$$\sum_{n=1}^N \sum_{t=0}^\infty \mathbb{E}_{\boldsymbol{w}_{1:t-1}}\left[\gamma^t \|\boldsymbol{\sigma}_{n-1}(\boldsymbol{x}'_t, \boldsymbol{\pi}(\boldsymbol{x}'_t))\|^2\right]$$

$$= \sum_{n=1}^N \sum_{t=0}^{T(n)-1} \mathbb{E}_{\boldsymbol{w}_{1:t-1}}\left[\gamma^t \|\boldsymbol{\sigma}_{n-1}(\boldsymbol{x}'_t, \boldsymbol{\pi}(\boldsymbol{x}'_t))\|^2\right] + \sum_{n=1}^N \sum_{t=T(n)}^\infty \mathbb{E}_{\boldsymbol{w}_{1:t-1}}\left[\gamma^t \|\boldsymbol{\sigma}_{n-1}(\boldsymbol{x}'_t, \boldsymbol{\pi}(\boldsymbol{x}'_t))\|^2\right]$$

$$\leq \sum_{n=1}^N \sum_{t=0}^{T(n)-1} \mathbb{E}_{\boldsymbol{w}_{1:t-1}}\left[\gamma^t \|\boldsymbol{\sigma}_{n-1}(\boldsymbol{x}'_t, \boldsymbol{\pi}(\boldsymbol{x}'_t))\|^2\right] + \sum_{n=1}^N \gamma^{T(n)} \frac{\sigma_{\max}^2}{1 - \gamma}$$

$$= \sum_{n=1}^N \sum_{t=0}^{T(n)-1} \mathbb{E}_{\boldsymbol{w}_{1:t-1}}\left[\gamma^t \|\boldsymbol{\sigma}_{n-1}(\boldsymbol{x}'_t, \boldsymbol{\pi}(\boldsymbol{x}'_t))\|^2\right] + \sum_{n=1}^N n^{-1} \frac{\sigma_{\max}^2}{1 - \gamma}$$

$$= \sum_{n=1}^{N} \sum_{t=0}^{T(n)-1} \mathbb{E}_{\boldsymbol{w}_{1:t-1}} \left[ \gamma^t \left\| \boldsymbol{\sigma}_{n-1}(\boldsymbol{x}'_t, \boldsymbol{\pi}(\boldsymbol{x}'_t)) \right\|^2 \right] + \frac{C\sigma_{\max}^2}{1-\gamma} \log(N)$$

Next, we bound the term

$$s_n = \sum_{t=0}^{T(n)-1} \gamma^t \sigma^{-2} \left\| \boldsymbol{\sigma}_{n-1}(\boldsymbol{x}'_t, \boldsymbol{\pi}(\boldsymbol{x}'_t)) \right\|^2.$$

Note that, $s_n \in \left[ 0, \frac{\sigma^{-2} d_x \sigma_{\max}^2}{1-\gamma} \right)$. Let $s_{\max} = \frac{\sigma^{-2} d_x \sigma_{\max}^2}{1-\gamma}$, we have $s_n \leq \frac{s_{\max}}{\log(1+s_{\max})} \log(1 + s_n)$ (Srinivas et al., 2012). Define $C_\gamma = \frac{s_{\max}}{\log(1+s_{\max})}$. We have,

$$s_n \leq C_\gamma \log \left( 1 + \sigma^{-2} \sum_{t=0}^{T(n)-1} \gamma^t \left\| \boldsymbol{\sigma}_{n-1}(\boldsymbol{x}'_t, \boldsymbol{\pi}(\boldsymbol{x}'_t)) \right\|^2 \right)$$

$$\leq C_\gamma \log \left( 1 + \sigma^{-2} \sum_{t=0}^{T(n)-1} \left\| \boldsymbol{\sigma}_{n-1}(\boldsymbol{x}'_t, \boldsymbol{\pi}(\boldsymbol{x}'_t)) \right\|^2 \right)$$

Finally, we have

$$\sum_{n=1}^{N} s_n \leq C_\gamma \sum_{n=1}^{N} \log \left( 1 + \sigma^{-2} \sum_{t=0}^{T(n)-1} \left\| \boldsymbol{\sigma}_{n-1}(\boldsymbol{x}'_t, \boldsymbol{\pi}(\boldsymbol{x}'_t)) \right\|^2 \right)$$

$$\leq C_\gamma \Gamma_{\sum_{n=1}^{N} T(n)} \qquad \text{(Corollary B.5)}$$

$$\leq C_\gamma \Gamma_{N \log(N)}$$

$$R_N = \sum_{n=1}^{N} r_n$$

$$\leq \sum_{n=1}^{N} (\lambda_n^2 + 2\lambda_n) \Sigma_n^\gamma(\boldsymbol{\pi}_n, \boldsymbol{f}^*)$$

$$\leq (\lambda_N^2 + 2\lambda_N) \sum_{n=1}^{N} \Sigma_n^\gamma(\boldsymbol{\pi}_n, \boldsymbol{f}^*)$$

$$= (\lambda_N^2 + 2\lambda_N) \sqrt{N} \sqrt{\sum_{n=1}^{N} (\Sigma_n^\gamma(\boldsymbol{\pi}_n, \boldsymbol{f}^*))^2}$$

$$\leq (2\lambda_N + \lambda_N^2) \sqrt{N} \times \sqrt{\sum_{n=1}^{N} \mathbb{E} \left[ \left( \sum_{t=0}^{\infty} \gamma^t \left\| \boldsymbol{\sigma}_n(\boldsymbol{x}_t, \boldsymbol{\pi}(\boldsymbol{x}_t)) \right\| \right)^2 \right]}$$

$$\leq (2\lambda_N + \lambda_N^2) \sqrt{N} \times \sqrt{\sum_{n=1}^{N} \mathbb{E} \left[ \left( \sum_{t=0}^{\infty} \gamma^t \right) \left( \sum_{t=0}^{\infty} \gamma^t \left\| \boldsymbol{\sigma}_n^2(\boldsymbol{x}_t, \boldsymbol{\pi}(\boldsymbol{x}_t)) \right\| \right) \right]}$$

$$= (2\lambda_N + \lambda_N^2) \sqrt{\frac{N}{1-\gamma}} \times \sqrt{\sum_{n=1}^{N} \mathbb{E} \left[ \sum_{t=0}^{\infty} \gamma^t \left\| \boldsymbol{\sigma}_n^2(\boldsymbol{x}_t, \boldsymbol{\pi}(\boldsymbol{x}_t)) \right\| \right]}$$

$$\leq (2\lambda_N + \lambda_N^2) \sqrt{\frac{C_\gamma N \Gamma_{N \log(N)}}{1-\gamma} + \frac{C\sigma_{\max}^2 N \log(N)}{(1-\gamma)^2}}$$

Since $\lambda_N \propto \beta_N / 1 - \gamma$, we get

$$R_N \leq \mathcal{O} \left( \Gamma_{N \log(N)}^{3/2} \sqrt{N} \right)$$

$\square$

## B.4 Analysis for the nonepisodic RL case

In this section, we prove Theorem 5.6. First, we restate the bounded energy assumption from Sukhija et al. (2024b).

**Definition B.8** ($\mathcal{K}_\infty$-functions). *The function $\xi : \mathbb{R}_{\geq 0} \to \mathbb{R}_{\geq 0}$ is of class $\mathcal{K}_\infty$, if it is continuous, strictly increasing, $\xi(0) = 0$ and $\xi(s) \to \infty$ for $s \to \infty$.*

**Assumption B.9** (Policies with bounded energy). *We assume there exists $\kappa, \xi \in \mathcal{K}_\infty$, positive constants $K, C_u, C_l$ with $C_u > C_l$, and $\gamma \in (0, 1)$ such that for each $\boldsymbol{\pi} \in \Pi$ we have,*

*Bounded energy:* There exists a Lyapunov function $V^{\boldsymbol{\pi}} : \mathcal{X} \to [0, \infty)$ for which $\forall \boldsymbol{x}, \boldsymbol{x}' \in \mathcal{X}$,

$$|V^{\boldsymbol{\pi}}(\boldsymbol{x}) - V^{\boldsymbol{\pi}}(\boldsymbol{x}')| \leq \kappa(\|\boldsymbol{x} - \boldsymbol{x}'\|) \qquad \text{(uniform continuity)}$$

$$C_l \xi(\|\boldsymbol{x}\|) \leq V^{\boldsymbol{\pi}}(\boldsymbol{x}) \leq C_u \xi(\|\boldsymbol{x}\|) \qquad \text{(positive definiteness)}$$

$$\mathbb{E}_{\boldsymbol{x}_+|\boldsymbol{x},\boldsymbol{\pi}}[V^{\boldsymbol{\pi}}(\boldsymbol{x}_+)] \leq \gamma V^{\boldsymbol{\pi}}(\boldsymbol{x}) + K \qquad \text{(drift condition)}$$

where $\boldsymbol{x}_+ = \boldsymbol{f}^*(\boldsymbol{x}, \boldsymbol{\pi}(\boldsymbol{x})) + \boldsymbol{w}$.

*Bounded norm of reward:*

$$\sup_{\boldsymbol{x} \in \mathcal{X}} \frac{r(\boldsymbol{x}, \boldsymbol{\pi}(\boldsymbol{x}))}{1 + V^{\boldsymbol{\pi}}(\boldsymbol{x})} < \infty$$

*Boundedness of the noise with respect to $\kappa$:*

$$\mathbb{E}_{\boldsymbol{w}}[\kappa(\|\boldsymbol{w}\|)] < \infty, \ \mathbb{E}_{\boldsymbol{w}}[\kappa^2(\|\boldsymbol{w}\|)] < \infty$$

Sukhija et al. (2024b) argue that this assumption is often satisfied in practice. We refer the reader to Sukhija et al. (2024b) for further details. Next, we make an assumption on the underlying system $\boldsymbol{f}^*$.

**Assumption B.10** (Continous closed-loop dynamics, and Gaussian noise.). *The dynamics model $\boldsymbol{f}^*$ and all $\boldsymbol{\pi} \in \Pi$ are continuous, and process noise is i.i.d. Gaussian with variance $\sigma^2$, i.e., $\boldsymbol{w}_t \overset{i.i.d}{\sim} \mathcal{N}(\boldsymbol{0}, \sigma^2 \boldsymbol{I})$.*

An important quantity in the average reward setting is the bias

$$B(\boldsymbol{\pi}, \boldsymbol{x}_0) = \lim_{T \to \infty} \mathbb{E}_{\boldsymbol{\pi}} \left[ \sum_{t=0}^{T-1} r(\boldsymbol{x}_t, \boldsymbol{u}_t) - J_{\text{avg}}(\boldsymbol{\pi}) \right]. \tag{16}$$

The Bellman equation for the average reward setting is given by

$$B(\boldsymbol{\pi}, \boldsymbol{x}) + J_{\text{avg}}(\boldsymbol{\pi}) = r(\boldsymbol{x}, \boldsymbol{\pi}(\boldsymbol{x})) + \mathbb{E}_{\boldsymbol{x}_+}[B(\boldsymbol{\pi}, \boldsymbol{x}_+)|\boldsymbol{x}, \boldsymbol{\pi}] \tag{17}$$

Sukhija et al. (2024b) show that under Assumption B.9 and Assumption B.10 the average reward solution and the bias (c.f., Equation (3)) are bounded. Moreover, they show that with Assumption 5.2 the average reward and bias are bounded for all dynamics $\boldsymbol{f} \in \mathcal{M}_n \cap \mathcal{M}_0$.

**Lemma B.11.** *Let Assumption 5.2, Assumption B.9, and Assumption 5.2 hold. Consider the following definitions*

$$J_{avg}(\boldsymbol{\pi}, \boldsymbol{f}^*) = \lim_{T \to \infty} \frac{1}{T} \mathbb{E} \left[ \sum_{t=0}^{T-1} r(\boldsymbol{x}_t, \boldsymbol{\pi}(\boldsymbol{x}_t)) \right] \ s.t., \ \boldsymbol{x}_{t+1} = \boldsymbol{f}^*(\boldsymbol{x}_t, \boldsymbol{\pi}(\boldsymbol{x}_t)) + \boldsymbol{w}_t, \quad \boldsymbol{x}_0 = \boldsymbol{x}(0),$$

$$J_{avg}(\boldsymbol{\pi}, \boldsymbol{f}) = \lim_{T \to \infty} \frac{1}{T} \mathbb{E} \left[ \sum_{t=0}^{T-1} r(\boldsymbol{x}'_t, \boldsymbol{\pi}(\boldsymbol{x}'_t)) \right] \ s.t., \ \boldsymbol{x}'_{t+1} = \boldsymbol{f}(\boldsymbol{x}'_t, \boldsymbol{\pi}(\boldsymbol{x}'_t)) + \boldsymbol{w}_t, \quad \boldsymbol{x}'_0 = \boldsymbol{x}(0),$$

$$\Sigma_n(\boldsymbol{\pi}, \boldsymbol{f}^*) = \lim_{T \to \infty} \frac{1}{T} \mathbb{E} \left[ \sum_{t=0}^{T-1} \|\boldsymbol{\sigma}_n(\boldsymbol{x}_t, \boldsymbol{\pi}(\boldsymbol{x}_t))\| \right] \ s.t., \ \boldsymbol{x}_{t+1} = \boldsymbol{f}^*(\boldsymbol{x}_t, \boldsymbol{\pi}(\boldsymbol{x}_t)) + \boldsymbol{w}_t, \quad \boldsymbol{x}_0 = \boldsymbol{x}(0),$$

$$\Sigma_n(\boldsymbol{\pi}, \boldsymbol{f}) = \lim_{T \to \infty} \frac{1}{T} \mathbb{E} \left[ \sum_{t=0}^{T-1} \|\boldsymbol{\sigma}_n(\boldsymbol{x}'_t, \boldsymbol{\pi}(\boldsymbol{x}'_t))\| \right] \ s.t., \ \boldsymbol{x}'_{t+1} = \boldsymbol{f}(\boldsymbol{x}'_t, \boldsymbol{\pi}(\boldsymbol{x}'_t)) + \boldsymbol{w}_t, \quad \boldsymbol{x}'_0 = \boldsymbol{x}(0),$$

$$\lambda_n = D_4(\boldsymbol{x}_0, \gamma, K)\beta_{n-1}(\delta),$$

and $D_4(\boldsymbol{x}_0, \gamma, K)$ is defined as in *Sukhija et al. (2024b, Theorem 3.1), is independent of n and increases with* $\|\boldsymbol{x}_0\|$, $K$ *and* $\gamma^{-1}$ *(see Sukhija et al. (2024b) for the exact dependence). Then we have for all* $n \geq 0$, $\boldsymbol{\pi} \in \Pi$, $\boldsymbol{f} \in \mathcal{M}_n \cap \mathcal{M}_0$ *with probability at least* $1 - \delta$

$$|J_{avg}(\boldsymbol{\pi}, \boldsymbol{f}^*) - J_{avg}(\boldsymbol{\pi}, \boldsymbol{f})| \leq \lambda_n \Sigma_n(\boldsymbol{\pi}, \boldsymbol{f})$$
$$|J_{avg}(\boldsymbol{\pi}, \boldsymbol{f}^*) - J_{avg}(\boldsymbol{\pi}, \boldsymbol{f})| \leq \lambda_n \Sigma_n(\boldsymbol{\pi}, \boldsymbol{f}^*)$$

*Proof.*

$$|J_{\text{avg}}(\boldsymbol{\pi}, \boldsymbol{f}) - J_{\text{avg}}(\boldsymbol{\pi}, \boldsymbol{f}^*)| = \left| \lim_{T \to \infty} \frac{1}{T} \mathbb{E}_{\boldsymbol{f}} \left[ \sum_{t=0}^{T-1} r(\boldsymbol{x}_t', \boldsymbol{\pi}(\boldsymbol{x}_t')) - J_{\text{avg}}(\boldsymbol{\pi}, \boldsymbol{f}^*) \right] \right|$$
$$= \left| \lim_{T \to \infty} \frac{1}{T} \mathbb{E}_{\boldsymbol{f}} \left[ \sum_{t=0}^{T-1} B(\boldsymbol{x}_t', \boldsymbol{\pi}(\boldsymbol{x}_t')) - B(\hat{\boldsymbol{x}}_{t+1}', \boldsymbol{\pi}(\hat{\boldsymbol{x}}_{t+1}')) \right] \right|$$
$$\leq \lambda_n \Sigma_n(\boldsymbol{\pi}, \boldsymbol{f}) \tag{1}$$

In the second last equality, we used the Bellman equation for the average reward setting (Equation (17)), where $\hat{\boldsymbol{x}}_{t+1}'$ is the next state under the true dynamics $\boldsymbol{f}^*$.

For the last inequality, Sukhija et al. (2024b) bound the bias term with $\lambda_n$ in Section A.3, on pages $23 - 24$.

We can use the same derivation to show that $|J_{\text{avg}}(\boldsymbol{\pi}, \boldsymbol{f}) - J_{\text{avg}}(\boldsymbol{\pi}, \boldsymbol{f}^*)| \leq \lambda_n \Sigma_n(\boldsymbol{\pi}, \boldsymbol{f}^*)$. $\square$

SOMBRL in the non-episodic setting operates similarly to NEORL (Sukhija et al., 2024b). In particular, we update our model and policy every $T_n$ step, where $T_n$ is defined as:

$$T_n = \max \left( \widehat{T_n}, \frac{\lceil \log (C_u/C_l) \rceil}{\log (1/\gamma)} \right), \tag{18}$$

$$\widehat{T_n} = \arg\max_{T \geq 1} T + 1 \tag{19}$$

$$\text{s.t.} \sum_{k=1}^{T} \sum_{j=1}^{d_x} \log \left( 1 + \sigma^{-2} \sigma_{n-1,j}^2(\boldsymbol{z}_{k,n}) \right) \leq \log(2). \tag{20}$$

Effectively, we update our model and policy only once we have accumulated more than one bit of information, i.e., $\sum_{k=1}^{T} \sum_{j=1}^{d_x} \log \left( 1 + \sigma^{-2} \sigma_{n-1,j}^2(\boldsymbol{z}_{k,n}) \right) > \log(2)$. With the updated model and model set $\mathcal{M}_n$, we select *any* dynamics in $\boldsymbol{f}_n \in \mathcal{M}_n \cap \mathcal{M}_0$ and pick the policy with

$$\boldsymbol{\pi}_n = \arg\max_{\boldsymbol{\pi} \in \Pi} J_{\text{avg}}(\boldsymbol{\pi}, \boldsymbol{f}_n) + \lambda_n \Sigma_n(\boldsymbol{\pi}, \boldsymbol{f}_n). \tag{21}$$

Note that (Sukhija et al., 2024b) require maximizing over the dynamics in $\mathcal{M}_n \cap \mathcal{M}_0$, whereas we do not. Moreover, while this optimization is generally intractable, for SOMBRL, we can obtain $\boldsymbol{f}$ using the quadratic program described in Equation (22). However, in practice, we just pick the mean model $\boldsymbol{\mu}_n \in \mathcal{M}_n$. This practical modification is also made in Sukhija et al. (2024b) where they optimize over dynamics in $\mathcal{M}_n$ instead of $\mathcal{M}_n \cap \mathcal{M}_0$.

**Theorem B.12** (Formal Theorem statement for informal Theorem 5.6)**.** *Define* $R_N = \sum_{n=1}^{N} \mathbb{E}[J_{avg}(\boldsymbol{\pi}^*) - r(\boldsymbol{x}_n, \boldsymbol{\pi}_n(\boldsymbol{x}_n)]$. *Let Assumption 5.2, Assumption B.9, and Assumption B.10 hold. Then we have for all* $N \geq 0$ *with probability at least* $1 - \delta$

$$R_N \leq \mathcal{O} \left( \Gamma_N^{3/2} \sqrt{N} \right)$$

*Proof.* Let $E_N$ denote the number of episodes after $N$ interactions in the environment.

$$R_N = \mathbb{E} \left[ \sum_{n=1}^{E_N} \sum_{k=0}^{T_n-1} J_{\text{avg}}(\boldsymbol{\pi}^*) - r(\boldsymbol{x}_k^n, \boldsymbol{\pi}_n(\boldsymbol{x}_k^n)) \right]$$
$$\leq \mathbb{E} \left[ \sum_{n=1}^{E_N} \sum_{k=0}^{T_n-1} J_{\text{avg}}(\boldsymbol{\pi}^*, \boldsymbol{f}_n) + \lambda_n \Sigma_n(\boldsymbol{\pi}^*, \boldsymbol{f}_n) - r(\boldsymbol{z}_k^n) \right] \qquad \text{(Lemma B.11)}$$

$$\leq \mathbb{E}\left[\sum_{n=1}^{E_N}\sum_{k=0}^{T_n-1} J_{\text{avg}}(\boldsymbol{\pi}_n, \boldsymbol{f}_n) + \lambda_n \Sigma_n(\boldsymbol{\pi}_n, \boldsymbol{f}_n) - r(\boldsymbol{z}_k^n)\right] \qquad \text{(Equation (21))}$$

$$\leq \mathbb{E}\left[\sum_{n=1}^{E_N}\sum_{k=0}^{T_n-1} J_{\text{avg}}(\boldsymbol{\pi}_n, \boldsymbol{f}_n) - r(\boldsymbol{z}_k^n)\right]$$

$$+ \lambda_N \mathbb{E}\left[\sum_{n=1}^{E_N}\sum_{k=0}^{T_n-1} \Sigma_n(\boldsymbol{\pi}_n, \boldsymbol{f}_n)\right]$$

$$\leq \mathcal{O}\left(\Gamma_N \sqrt{N}\right) + \mathbb{E}\left[\lambda_N \sum_{n=1}^{E_N}\sum_{k=0}^{T_n-1} \Sigma_n(\boldsymbol{\pi}_n, \boldsymbol{f})\right] \qquad \text{(Theorem 3.1 Sukhija et al. (2024b))}$$

Next, we focus on $\mathbb{E}\left[\lambda_N \sum_{n=1}^{E_N}\sum_{k=0}^{T_n-1} \Sigma_n(\boldsymbol{\pi}_n, \boldsymbol{f})\right]$

$$\mathbb{E}\left[\sum_{n=1}^{E_N}\sum_{k=0}^{T_n-1} \Sigma_n(\boldsymbol{\pi}_n, \boldsymbol{f})\right]$$

$$= \mathbb{E}\left[\sum_{n=1}^{E_N}\sum_{k=0}^{T_n-1} \|\boldsymbol{\sigma}_n(\boldsymbol{x}_t, \boldsymbol{\pi}(\boldsymbol{x}_t))\|\right]$$

$$+ \mathbb{E}\left[\sum_{n=1}^{E_N}\sum_{k=0}^{T_n-1} \Sigma_n(\boldsymbol{\pi}_n, \boldsymbol{f}) - \|\boldsymbol{\sigma}_n(\boldsymbol{x}_t, \boldsymbol{\pi}(\boldsymbol{x}_t))\|\right]$$

$$\leq C\sqrt{\Gamma_N N} + \mathbb{E}\left[\sum_{n=1}^{E_N}\sum_{k=0}^{T_n-1} \Sigma_n(\boldsymbol{\pi}_n, \boldsymbol{f}) - \|\boldsymbol{\sigma}_n(\boldsymbol{x}_t, \boldsymbol{\pi}(\boldsymbol{x}_t))\|\right]$$
$$\text{(Lemma A.1 Sukhija et al. (2024b))}$$

$$\leq \sqrt{N\Gamma_N} + \mathcal{O}\left(\Gamma_N \sqrt{N}\right) \qquad \text{(Sukhija et al. (2024b, Theorem 3.1) with reward } \boldsymbol{\sigma}_n)$$

Therefore

$$\mathbb{E}\left[\lambda_T \sum_{n=1}^{E_N}\sum_{k=0}^{T_n-1} \Sigma_n(\boldsymbol{\pi}_n, \boldsymbol{f})\right] \leq \mathcal{O}\left(\lambda_N \Gamma_N \sqrt{N}\right)$$
$$\leq \mathcal{O}\left(\Gamma_N^{3/2}\sqrt{N}\right)$$

In conclusion,

$$R_N \leq \mathcal{O}\left(\Gamma_N^{3/2}\sqrt{N}\right)$$

$\square$

## B.5 Analysis for pure intrinsic exploration

In the following, we derive a sample complexity bound for a pure intrinsic exploration algorithm. Thereby showing convergence for methods such Buisson-Fenet et al. (2020).

**Theorem B.13.** *Let Assumption 5.1 and Assumption 5.2 hold. Consider* SOMBRL *with extrinsic reward $r = 0$, i.e.,*

$$\boldsymbol{\pi}_n = \arg\max_{\boldsymbol{\pi}\in\Pi} \mathbb{E}_{\boldsymbol{\pi}}\left[\sum_{t=0}^{T-1} \|\boldsymbol{\sigma}_n(\boldsymbol{x}_t', \boldsymbol{u}_t)\|\right], \quad \boldsymbol{x}_{t+1}' = \boldsymbol{\mu}_n(\boldsymbol{x}_t', \boldsymbol{u}_t) + \boldsymbol{w}_t.$$

*Then we have $\forall N > 0$, with probability at least $1 - \delta$*

$$\max_{\boldsymbol{\pi}\in\Pi} \mathbb{E}_{\boldsymbol{f}^*}\left[\sum_{t=0}^{T-1} \|\boldsymbol{\sigma}_n(\boldsymbol{x}_t, \boldsymbol{\pi}(\boldsymbol{x}_t))\|\right] \leq \mathcal{O}\left(\sqrt{\frac{\Gamma_N^3}{N}}\right).$$

*Proof.* Let $\Sigma_N^* = \max_{\boldsymbol{\pi}} \Sigma_N(\boldsymbol{\pi}, \boldsymbol{f}^*)$ and $\boldsymbol{\pi}_N^*$ the corresponding policy.

$$\Sigma_N^* \leq \frac{1}{N} \sum_{n=1}^{N} \Sigma_n^* \qquad \text{(monotoncity of the variance)}$$

$$\leq \frac{1}{N} \sum_{n=1}^{N} (1 + \lambda_n) \Sigma_n(\boldsymbol{\pi}_n^*, \boldsymbol{\mu}_n) \qquad \text{(Lemma B.3)}$$

$$\leq \frac{1}{N} \sum_{n=1}^{N} (1 + \lambda_n) \Sigma_n(\boldsymbol{\pi}_n, \boldsymbol{\mu}_n) \qquad (\boldsymbol{\pi}_n \text{ is the maximizer for mean dynamics } \boldsymbol{\mu}_n)$$

$$\leq \frac{1}{N} \sum_{n=1}^{N} (1 + \lambda_n)^2 \Sigma_n(\boldsymbol{\pi}_n, \boldsymbol{f}^*) \qquad \text{(Lemma B.3)}$$

$$\leq (1 + \lambda_N)^2 \frac{1}{N} \sum_{n=1}^{N} \Sigma_n(\boldsymbol{\pi}_n, \boldsymbol{f}^*)$$

$$\leq (1 + \lambda_N)^2 \frac{1}{\sqrt{N}} \sum_{n=1}^{N} \Sigma_n^2(\boldsymbol{\pi}_n, \boldsymbol{f}^*)$$

$$\leq \mathcal{O}\left(\sqrt{\frac{\Gamma_N^3}{N}}\right)$$

$\square$

Effectively, Theorem B.13 shows that pure intrinsic exploration reduces our model epistemic uncertainty with a rate of $\sqrt{\frac{\Gamma_N^3}{N}}$. To the best of our knowledge, we are the first to show this. Moreover, Sukhija et al. (2024c) derive a similar bound but their algorithm performs optimistic exploration from Equation (7) in addition to maximizing the intrinsic rewards. Our result shows that the optimistic exploration is not necessary for this setting.

### B.6 Analysis for the finite horizon setting with sub-Gaussia noise

In the following, we analyse the regret for the setting where the process noise $\boldsymbol{w}$ is $\sigma$-sub Gaussian.
**Assumption B.14.** *The dynamics model $\boldsymbol{f}^*$, reward $r$, and all $\boldsymbol{\pi} \in \Pi$ are $L_f$, $L_r$ and $L_{\boldsymbol{\pi}}$ Lipschitz, respectively. Furthermore, we assume that process noise is i.i.d. $\sigma$-sub Gaussian.*

We make the same assumptions as other works (Curi et al., 2020; Sussex et al., 2023) that study this setting. Moreover, Lipschitz continuity is a common assumption for nonlinear dynamics (Khalil, 2015) and is satisfied for many real-world systems.

Curi et al. (2020) provide a regret bound that depends exponentially on the horizon $T$, i.e., $R_N \in \mathcal{O}\left(\sqrt{\Gamma_N^T N}\right)$. They obtain an exponential dependence because when planning optimistically, i.e., solving Equation (7), they consider all plausible dynamics, including those that are not Lipschitz continuous for all $n$. Solving Equation (7) for only continuous dynamics is intractable. However, for SOMBRL, as we do not maximize over the set of dynamics we can overcome this limitation.

Moreover, since $\boldsymbol{f}^*$ has bounded RKHS norm, i.e., $\|\boldsymbol{f}^*\|_k \leq B$ ( Assumption 5.2). From Srinivas et al. (2012); Chowdhury & Gopalan (2017) follows that with probability $1 - \delta$ we have for every $n$:

$$\|\boldsymbol{f}^* - \boldsymbol{\mu}_n\|_{k_n} \leq \beta_n.$$

For SOMBRL, instead of planning with the mean, which in general might not be Lipschitz continuous for all $n$, we select a function $\boldsymbol{f}_n$ that not only approximates the $\boldsymbol{f}^*$ function well, i.e., $\|\boldsymbol{f}^* - \boldsymbol{f}_n\|_{k_n} \leq \beta_n$, but also its RKHS norm does not grow with $n$. To do that we propose to solve the following quadratic optimization problem:

$$\boldsymbol{f}_n = \underset{\boldsymbol{f} \in \text{span}(k(\boldsymbol{x}_1, \cdot), \dots, k(\boldsymbol{x}_n, \cdot))}{\arg\min} \|\boldsymbol{f} - \boldsymbol{\mu}_n\|_{k_n} \tag{22}$$
$$\text{s.t. } \|\boldsymbol{f}\|_k \leq B$$

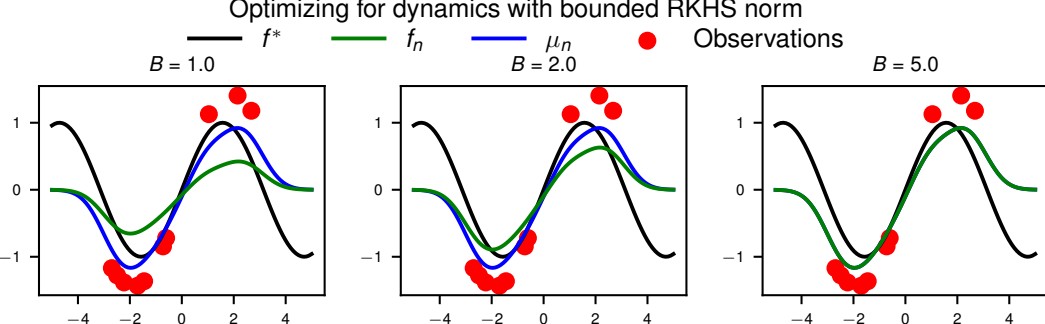

Figure 5: Solution to Equation (22) for different values for $B$. Effectively, for larger values for $B$, $\boldsymbol{\mu}_n$ and $\boldsymbol{f}_n$ coincide.

**Theorem B.15.** *The optimization problem Equation* (22) *is feasible and we have* $\|\boldsymbol{f}_n - \boldsymbol{\mu}_n\|_{k_n} \le \beta_n$.

*Proof.* Consider the noise-free case, i.e., $\boldsymbol{w} = 0$, and let $\bar{\boldsymbol{\mu}}_n$ posterior mean for this setting. For the function $\bar{\boldsymbol{\mu}}_n$ holds that $\|\boldsymbol{f}^* - \bar{\boldsymbol{\mu}}_n\|_{k_n} \le \beta_n$ (Corollary 3.11 of Kanagawa et al. (2018)) and $\|\bar{\boldsymbol{\mu}}_n\|_k \le B$ (Theorem 3.5 of Kanagawa et al. (2018)). Since $\|\bar{\boldsymbol{\mu}}_n - \boldsymbol{\mu}_n\|_{k_n} \le \|\bar{\boldsymbol{\mu}}_n - \boldsymbol{f}^*\|_{k_n} + \|\boldsymbol{f}^* - \boldsymbol{\mu}_n\|_{k_n} \le 2\beta_n$. By representer theorem, it also holds that $\bar{\boldsymbol{\mu}}_n \in \text{span}(k(\boldsymbol{z}_1, \cdot), \ldots, k(\boldsymbol{z}_n, \cdot))$. $\qquad\square$

Let $\boldsymbol{\alpha}_n = (\boldsymbol{K} + \sigma^2 \boldsymbol{I})^{-1} \boldsymbol{y} \in \mathbb{R}^n$ and reparametrize $\boldsymbol{f}(\boldsymbol{x}) = \sum_{i=1}^{n} \alpha_i k(\boldsymbol{x}_i, \boldsymbol{x})$. We have $\|\boldsymbol{f}\|_k^2 = \boldsymbol{\alpha}^\top \boldsymbol{K} \boldsymbol{\alpha}$. We also have:

$$\|\boldsymbol{f} - \boldsymbol{\mu}_n\|_{k_n}^2 = (\boldsymbol{\alpha} - \boldsymbol{\alpha}_n)^\top \boldsymbol{K} \left( \boldsymbol{I} + \frac{1}{\sigma^2} \boldsymbol{K} \right) (\boldsymbol{\alpha} - \boldsymbol{\alpha}_n)$$

Hence the optimization problem Equation (22) is equivalent to:

$$\min_{\boldsymbol{\alpha} \in \mathbb{R}^n} (\boldsymbol{\alpha} - \boldsymbol{\alpha}_n)^\top \boldsymbol{K} \left( \boldsymbol{I} + \frac{1}{\sigma^2} \boldsymbol{K} \right) (\boldsymbol{\alpha} - \boldsymbol{\alpha}_n)$$

$$\text{s.t. } \boldsymbol{\alpha}^\top \boldsymbol{K} \boldsymbol{\alpha} \le B^2$$

This is a quadratic program that can be solved using any QP solver. The program finds the closest function to the posterior mean $\boldsymbol{\mu}_n$ that is Lipschitz continuous (see Figure 5). In particular, note that since $\|\boldsymbol{f}_n\|_k \le B$, $\boldsymbol{f}_n$ has a Lipschitz constant $L_B$ which is independent of $n$ (Berkenkamp, 2019). From hereon, let $L_* = \max\{L_f, L_B\}$.

For the sub-Gaussian case, SOMBRL follows the same strategy as Equation (8) but instead of using the mean dynamics $\boldsymbol{\mu}_n$, we plan with the dynamics $\boldsymbol{f}_n$ that are obtained from solving Equation (22).

$$\boldsymbol{\pi}_n = \arg\max_{\boldsymbol{\pi} \in \Pi} \mathbb{E}_{\boldsymbol{\pi}} \left[ \sum_{t=0}^{T-1} r(\boldsymbol{x}_t', \boldsymbol{u}_t) + \lambda_n \|\boldsymbol{\sigma}_n(\boldsymbol{x}_t', \boldsymbol{u}_t)\| \right] \qquad (23)$$

$$\boldsymbol{x}_{t+1}' = \boldsymbol{f}_n(\boldsymbol{x}_t', \boldsymbol{u}_t) + \boldsymbol{w}_t,$$

**Lemma B.16.** *Let Assumption B.14 and Assumption 5.2 hold. Consider the following definitions*

$$J(\boldsymbol{\pi}, \boldsymbol{f}^*) = E \left[ \sum_{t=0}^{T-1} r(\boldsymbol{x}_t, \boldsymbol{\pi}(\boldsymbol{x}_t)) \right] \quad s.t., \ \boldsymbol{x}_{t+1} = \boldsymbol{f}^*(\boldsymbol{x}_t, \boldsymbol{\pi}(\boldsymbol{x}_t)) + \boldsymbol{w}_t, \quad \boldsymbol{x}_0 = \boldsymbol{x}(0),$$

$$J(\boldsymbol{\pi}, \boldsymbol{f}_n) = E \left[ \sum_{t=0}^{T-1} r(\boldsymbol{x}_t', \boldsymbol{\pi}(\boldsymbol{x}_t')) \right] \quad s.t., \ \boldsymbol{x}_{t+1}' = \boldsymbol{f}_n(\boldsymbol{x}_t', \boldsymbol{\pi}(\boldsymbol{x}_t')) + \boldsymbol{w}_t, \quad \boldsymbol{x}_0' = \boldsymbol{x}(0),$$

$$\Sigma_n(\boldsymbol{\pi}, \boldsymbol{f}^*) = E \left[ \sum_{t=0}^{T-1} \|\boldsymbol{\sigma}_n(\boldsymbol{x}_t, \boldsymbol{\pi}(\boldsymbol{x}_t))\| \right] \quad s.t., \ \boldsymbol{x}_{t+1} = \boldsymbol{f}^*(\boldsymbol{x}_t, \boldsymbol{\pi}(\boldsymbol{x}_t)) + \boldsymbol{w}_t, \quad \boldsymbol{x}_0 = \boldsymbol{x}(0),$$

$$\Sigma_n(\boldsymbol{\pi}, \boldsymbol{f}_n) = E \left[ \sum_{t=0}^{T-1} \|\boldsymbol{\sigma}_n(\boldsymbol{x}_t', \boldsymbol{\pi}(\boldsymbol{x}_t'))\| \right] \quad s.t., \ \boldsymbol{x}_{t+1}' = \boldsymbol{f}_n(\boldsymbol{x}_t', \boldsymbol{\pi}(\boldsymbol{x}_t')) + \boldsymbol{w}_t, \quad \boldsymbol{x}_0' = \boldsymbol{x}(0),$$

$$\lambda_n = (1 + d_x)L_r(1 + L_{\boldsymbol{\pi}})\bar{L}_*^{T-1}T\beta_n.$$

*Then we have for all $n \geq 0$, $\boldsymbol{\pi} \in \Pi$ with probability at least $1 - \delta$*

$$|J(\boldsymbol{\pi}, \boldsymbol{f}^*) - J(\boldsymbol{\pi}, \boldsymbol{f}_n)| \leq \lambda_n \Sigma_n(\boldsymbol{\pi}, \boldsymbol{f}_n)$$
$$|J(\boldsymbol{\pi}, \boldsymbol{f}^*) - J(\boldsymbol{\pi}, \boldsymbol{f}_n)| \leq \lambda_n \Sigma_n(\boldsymbol{\pi}, \boldsymbol{f}^*)$$

*Proof.*

$$|J(\boldsymbol{\pi}, \boldsymbol{f}^*) - J(\boldsymbol{\pi}, \boldsymbol{f}_n)| = \mathbb{E}\left[\sum_{t=0}^{T-1} r(\boldsymbol{x}_t, \boldsymbol{\pi}(\boldsymbol{x}_t)) - r(\boldsymbol{x}_t', \boldsymbol{\pi}(\boldsymbol{x}_t'))\right] \leq L_r(1 + L_{\boldsymbol{\pi}})\mathbb{E}\left[\sum_{t=0}^{T-1} \|\boldsymbol{x}_t - \boldsymbol{x}_t'\|\right]$$

Next we analyze $\|\boldsymbol{x}_t - \boldsymbol{x}_t'\|$ for any $t$. Without loss of generality, assume $L_* \geq 1$ and define $\bar{L}_* = L_*(1 + L_{\boldsymbol{\pi}})$.

We show that

$$\left\|\boldsymbol{x}_{t+1} - \boldsymbol{x}_{t+1}'\right\| \leq (1 + \sqrt{d_x})\beta_n \left(\sum_{k=0}^{t} \bar{L}_*^{t-k} \|\boldsymbol{\sigma}_n(\boldsymbol{x}_k', \boldsymbol{\pi}(\boldsymbol{x}_k'))\|\right).$$

Consider $t = 1$

$$\|\boldsymbol{x}_1 - \boldsymbol{x}_1'\| = \|\boldsymbol{f}^*(\boldsymbol{x}_0', \boldsymbol{\pi}(\boldsymbol{x}_0')) - \boldsymbol{f}_n(\boldsymbol{x}_0', \boldsymbol{\pi}(\boldsymbol{x}_0'))\|$$
$$\leq (1 + \sqrt{d_x})\beta_n \|\boldsymbol{\sigma}_n(\boldsymbol{x}_0', \boldsymbol{\pi}(\boldsymbol{x}_0'))\|$$

Consider any $t > 1$,

$$\left\|\boldsymbol{x}_{t+1} - \boldsymbol{x}_{t+1}'\right\| = \|\boldsymbol{f}^*(\boldsymbol{x}_t, \boldsymbol{\pi}(\boldsymbol{x}_t)) - \boldsymbol{f}_n(\boldsymbol{x}_t', \boldsymbol{\pi}(\boldsymbol{x}_t'))\|$$
$$\leq \|\boldsymbol{f}^*(\boldsymbol{x}_t', \boldsymbol{\pi}(\boldsymbol{x}_t')) - \boldsymbol{f}_n(\boldsymbol{x}_t', \boldsymbol{\pi}(\boldsymbol{x}_t'))\| + \|\boldsymbol{f}^*(\boldsymbol{x}_t, \boldsymbol{\pi}(\boldsymbol{x}_t)) - \boldsymbol{f}^*(\boldsymbol{x}_t', \boldsymbol{\pi}(\boldsymbol{x}_t'))\|$$
$$\leq (1 + \sqrt{d_x})\beta_n \|\boldsymbol{\sigma}_n(\boldsymbol{x}_t', \boldsymbol{\pi}(\boldsymbol{x}_t'))\| + \bar{L}_* \|\boldsymbol{x}_t - \boldsymbol{x}_t'\|$$
$$\leq (1 + \sqrt{d_x})\beta_n (\|\boldsymbol{\sigma}_n(\boldsymbol{x}_t', \boldsymbol{\pi}(\boldsymbol{x}_t'))\|) + (1 + \sqrt{d_x})\beta_n \left(\bar{L}_* \left(\sum_{k=0}^{t-1} \bar{L}_*^{t-1-k} \|\boldsymbol{\sigma}_n(\boldsymbol{x}_k', \boldsymbol{\pi}(\boldsymbol{x}_k'))\|\right)\right)$$
$$= (1 + \sqrt{d_x})\beta_n \left(\sum_{k=0}^{t} \bar{L}_*^{t-k} \|\boldsymbol{\sigma}_n(\boldsymbol{x}_k', \boldsymbol{\pi}(\boldsymbol{x}_k'))\|\right)$$

In particular, since $\bar{L}_* \geq 1$, we have $\left\|\boldsymbol{x}_{t+1} - \boldsymbol{x}_{t+1}'\right\| \leq (1 + \sqrt{d_x})\beta_n \bar{L}_*^t \left(\sum_{k=0}^{t-1} \|\boldsymbol{\sigma}_n(\boldsymbol{x}_k', \boldsymbol{\pi}(\boldsymbol{x}_k'))\|\right)$.

In summary, we have

$$|J(\boldsymbol{\pi}, \boldsymbol{f}^*) - J(\boldsymbol{\pi}, \boldsymbol{\mu}_n)| = \mathbb{E}\left[\sum_{t=0}^{T-1} r(\boldsymbol{x}_t, \boldsymbol{\pi}(\boldsymbol{x}_t)) - r(\boldsymbol{x}_t', \boldsymbol{\pi}(\boldsymbol{x}_t'))\right]$$
$$\leq L_r(1 + L_{\boldsymbol{\pi}})\mathbb{E}\left[\sum_{t=0}^{T-1} \|\boldsymbol{x}_t - \boldsymbol{x}_t'\|\right]$$
$$\leq L_r(1 + L_{\boldsymbol{\pi}})(1 + \sqrt{d_x}) \times \mathbb{E}\left[\sum_{t=0}^{T-1} \beta_n \bar{L}_*^{t-1} \left(\sum_{k=0}^{t-1} \|\boldsymbol{\sigma}_n(\boldsymbol{x}_k', \boldsymbol{\pi}(\boldsymbol{x}_k'))\|\right)\right]$$
$$\leq (1 + d_x)L_r(1 + L_{\boldsymbol{\pi}})\bar{L}_*^{T-1}T\beta_n\Sigma_n(\boldsymbol{\pi}, \boldsymbol{\mu}_n)$$
$$= \lambda_n\Sigma_n(\boldsymbol{\pi}, \boldsymbol{\mu}_n).$$

$\square$

The main difference between our analysis and the analysis from Curi et al. (2020) is that for us $\lambda_n \propto \beta_n$ if we plan with $\boldsymbol{f}_n$.

**Lemma B.17.** *Let Assumption B.14 and Assumption 5.2 hold and consider the simple regret at episode $n$, $r_n = J(\boldsymbol{\pi}^*, \boldsymbol{f}^*) - J(\boldsymbol{\pi}_n, \boldsymbol{f}^*)$. The following holds for all $n > 0$ with probability at least $1 - \delta$*

$$r_n \leq (2\lambda_n + \lambda_n^2)\Sigma_n(\boldsymbol{\pi}_n, \boldsymbol{f}^*)$$

*Proof.*

$$
\begin{aligned}
r_n &= J(\boldsymbol{\pi}^*, \boldsymbol{f}^*) - J(\boldsymbol{\pi}_n, \boldsymbol{f}^*) \\
&\leq J(\boldsymbol{\pi}^*, \boldsymbol{f}_n) + \lambda_n \Sigma_n(\boldsymbol{\pi}^*, \boldsymbol{f}_n) - J(\boldsymbol{\pi}_n, \boldsymbol{f}^*) && \text{(Lemma B.16)} \\
&\leq J(\boldsymbol{\pi}_n, \boldsymbol{f}_n) + \lambda_n \Sigma_n(\boldsymbol{\pi}_n, \boldsymbol{f}_n) - J(\boldsymbol{\pi}_n, \boldsymbol{f}^*) && \text{(Equation (23))} \\
&= J(\boldsymbol{\pi}_n, \boldsymbol{f}_n) - J(\boldsymbol{\pi}_n, \boldsymbol{f}^*) + \lambda_n \Sigma_n(\boldsymbol{\pi}_n, \boldsymbol{f}_n) \\
&\leq \lambda_n \Sigma_n(\boldsymbol{\pi}_n, \boldsymbol{f}^*) + \lambda_n \Sigma_n(\boldsymbol{\pi}_n, \boldsymbol{f}_n) && \text{(Lemma B.16)} \\
&= 2\lambda_n \Sigma_n(\boldsymbol{\pi}_n, \boldsymbol{f}^*) + \lambda_n (\Sigma_n(\boldsymbol{\pi}_n, \boldsymbol{f}_n) - \Sigma_n(\boldsymbol{\pi}_n, \boldsymbol{f}^*)) \\
&\leq (\lambda_n^2 + 2\lambda_n)\Sigma_n(\boldsymbol{\pi}_n, \boldsymbol{f}^*).
\end{aligned}
$$

$\square$

**Theorem B.18** (Finite horizon setting sub-Gaussian case)**.** *Let Assumption B.14 and Assumption 5.2 hold. Then we have $\forall N > 0$ with probability at least $1 - \delta$*

$$R_N \leq \mathcal{O}\left(\Gamma_N^{3/2}\sqrt{N}\right).$$

*Proof.* The proof is the same as for Theorem 5.4, since in Lemma B.17 we show that also for the sub-Gaussian case, SOMBRL has the same regret dependence w.r.t. $\lambda_n$ and $\Sigma_n(\boldsymbol{\pi}_n, \boldsymbol{f}^*)$. $\square$

# C Additional Experiments

In this section, we provide additional experiments.

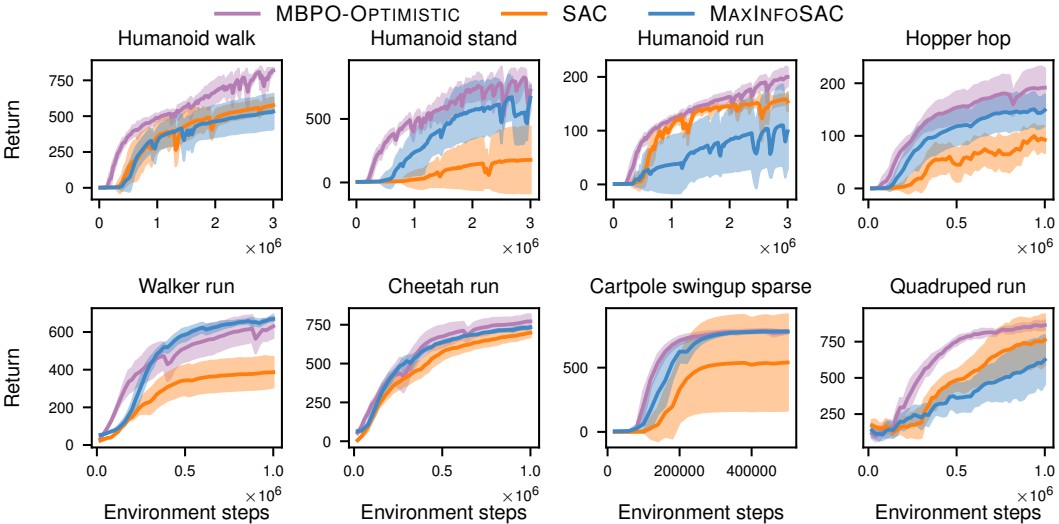

Figure 6: Comparison between MBPO-OPTIMSTIC and MAXINFOSAC and SAC. We observe that MBPO-OPTIMSTIC, being an MBRL algorithm, performs the best in terms of sample efficiency.

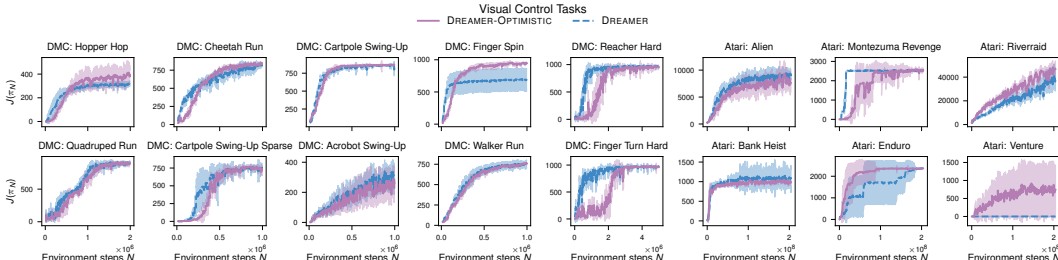

Figure 7: Learning curves for the visual control tasks from DMC and Atari using DREAMER as the base algorithm. DREAMER-OPTIMISTIC either performs on-par or better than DREAMER in all our experiments. Particularly, in the Venture task from the Atari benchmark, where DREAMER fails to obtain any rewards.

**Experiments with MBPO** In Figure 6 we compare MBPO-OPTIMISTIC with off-policy RL algorithms MAXINFOSAC (Sukhija et al., 2024a) and SAC (Haarnoja et al., 2018). From the figure, we conclude that MBPO-OPTIMISTIC performs the best in terms of sample-efficiency, particularly for the challenging/high-dimensional humanoid tasks. Moreover, between SAC and MAXINFOSAC, the latter achieves much better performance. We believe this is due to its intrinsic exploration reward.

**Experiments with DREAMER** In Figure 7 we compare DREAMER-OPTIMISTIC with DREAMER on additional environments. Overall, we observe that DREAMER-OPTIMISTIC performs either on par or better than DREAMER. However, for certain environments such as Reacher Hard or Finger Turn Hard, DREAMER is more sample-efficient. We believe this is because in these settings smaller values for $\lambda_n$ would suffice for exploration. However, we use a constant value for $\lambda_n$ across all environments and automatically update it using the approach proposed in Sukhija et al. (2024a). Investigating alternative strategies for $\lambda_n$, would generally benefit SOMBRL methods. We think this is a promising direction for future work.

In Figure 8 and Figure 9 we compare DREAMER-OPTIMISTIC with DREAMER on proprioceptive tasks. In most environments, DREAMER-OPTIMISTIC performs on par. It performs better in the Finger Spin environment. However, when action costs are introduced (Figure 9), in line with our results in Section 6, DREAMER fails to obtain any meaningful rewards.

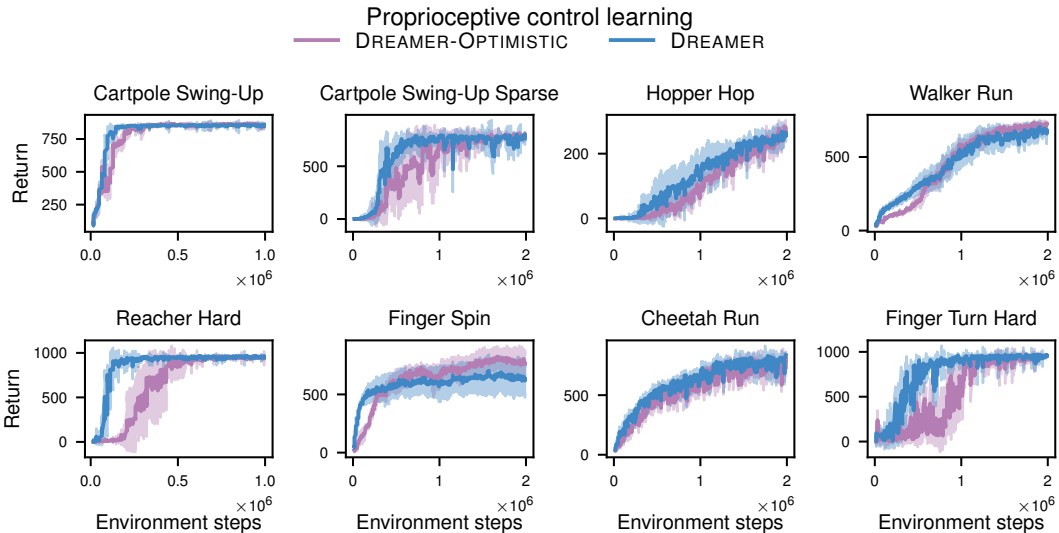

Figure 8: Experiments with DREAMER-OPTIMISTIC and DREAMER for proprioceptive tasks. DREAMER-OPTIMISTIC performs on par with DREAMER, obtaining slightly better performance on the Finger Spin task.

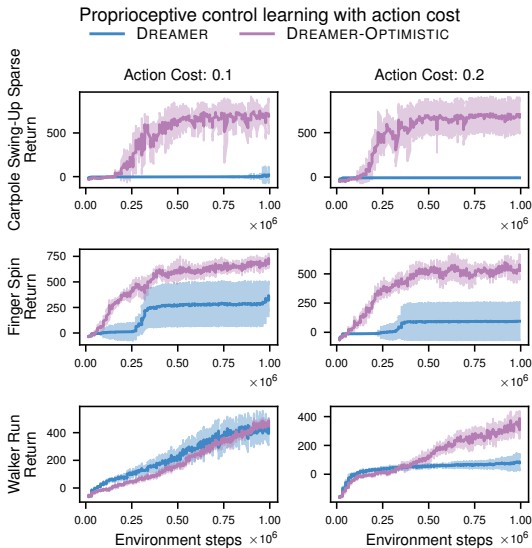

Figure 9: Experiments with DREAMER-OPTIMISTIC and DREAMER for proprioceptive tasks with action costs. DREAMER completely fails to solve the task, whereas DREAMER-OPTIMISTIC does not.

# D Experiment Details

In this section, we provide additional details for our experiments.

## D.1 MBPO-OPTIMISTIC

For MBPO-OPTIMISTIC, we train an ensemble of forward dynamics models[6]. We use the disagreement between the ensembles to quantify model epistemic uncertainty, similar to Pathak et al. (2019); Curi et al. (2020); Sukhija et al. (2024c). For selecting $\lambda_n$, we use the auto-tuning approach from Sukhija et al. (2024a), where the intrinsic reward weight is optimized by minimizing the following loss with stochastic gradient descent

$$L(\lambda) = \mathop{\mathbb{E}}_{\boldsymbol{x}\sim\mathcal{D}_{1:n},\boldsymbol{u}\sim\boldsymbol{\pi}_n,\bar{\boldsymbol{u}}\sim\bar{\boldsymbol{\pi}}_n} \log(\lambda)(\boldsymbol{\sigma}_n(\boldsymbol{x},\boldsymbol{u}) - \boldsymbol{\sigma}_n(\boldsymbol{x},\bar{\boldsymbol{u}})). \tag{24}$$

Here $\bar{\boldsymbol{\pi}}_n$ is a target policy, which is updated using polyak updates of $\boldsymbol{\pi}_n$. This objective increases $\lambda$ when the policy is under exploring compared to the target policy. Sukhija et al. (2024a) show that this strategy works across several model-free off-policy RL algorithms.

Besides using the model to quantify disagreement, we generate additional data by adding the transitions predicted by our learned model. In particular, for every policy update, we sample a batch of transitions from the data buffer $(\boldsymbol{x},\boldsymbol{u},\boldsymbol{x}') \sim \mathcal{D}_{1:n}$, and add $(\boldsymbol{x},\boldsymbol{u},\hat{\boldsymbol{x}}')$, transitions predicted by our mean model $\boldsymbol{\mu}_n$, to the batch. This allows us to combine true rollouts with model generated rollouts, as proposed in Janner et al. (2019). Since we can generate additional data through our learned model, we can efficiently increase our update-to-data ratio (UTD). For all our experiments with MBPO, with use an UTD of $5$[7].

We use the same hyperparameters as Sukhija et al. (2024a) for all our state-based experiments.

## D.2 DREAMER-OPTIMISTIC

We use DREAMERV3 As the base model. For quantifying the model epistemic uncertainty, we use the same approach as Sekar et al. (2020); Mendonca et al. (2021) and learn an ensemble of MLPs to model the latent dynamics[8]. The ensemble is only used for quantifying the model uncertainty/intrinsic reward. For the policy optimization, we use the DREAMER backbone, where the agent optimizes the policy using imagined rollouts. For selecting $\lambda$, we also use the objective in Equation (24). We found adding a regularize term $\alpha * |\lambda|$ to the objective worked better with DREAMER. We initialize $\lambda$ with 2 and pick $\alpha = 0.001$. For the rest, we use the same hyperparemters as DREAMER[9].

## D.3 SIMFSVGD-OPTIMISTIC

### Tolerance reward with different margins

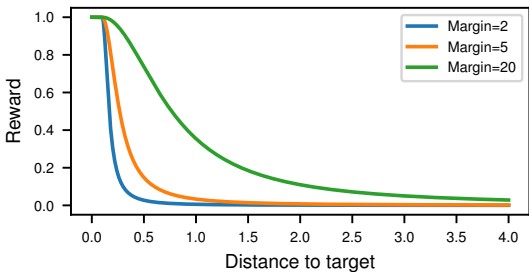

Figure 10: Tolerance reward function for different values of the margin. For larger margins, the agent receives rewards even if its further away from the target.

We use the same experiment setup, simulation prior, and hyperparameters as Rothfuss et al. (2024)[10]. The reward function in Rothfuss et al. (2024) is based on the tolerance reward from Tassa et al. (2018).

---

[6]For all tasks we use a $(256, 256)$ neural network architecture with 5 ensembles, except for the humanoid and quadruped tasks where we use $(512, 512)$.

[7]We did not tune the UTD and chose 5 to trade-off between computational cost and sample efficiency.

[8]For all tasks we use a $(512, 512)$ neural network architecture with 5 ensembles.

[9]We use the 12 million size model and the official DREAMERV3 implementation (`https://github.com/danijar/dreamerv3/tree/main`).

[10]official implementation: `https://github.com/lasgroup/simulation_transfer`

The tolerance function, gives higher rewards when the agent is close to a desired state, i.e., in case of the RC car the target position. The "closesness" is quantified using a margin parameter for the reward function. In Figure 10 we plot the reward for different margin parameters. As we decrease the margin, the reward becomes sparser. Rothfuss et al. (2024) use a margin of 20. In our simulation experiments, we show that SIMFSVGD performs worse than SIMFSVGD-OPTIMISTIC for smaller margins. For our hardware experiment, we use a margin of 5, for which SIMFSVGD fails to learn. For $\lambda_n$ we found that a linearly decaying schedule worked the best. Therefore, we linearly interpolated from $\lambda_0 = 0.5$ and $\lambda_{10} = 0$. After the tenth episode, the agent greedily maximized the extrinsic reward.

### D.4  GP experiments

For our GP experiments, we use the RBF kernel. The kernel parameters are updated online using maximum likelihood estimation (Rasmussen & Williams, 2005). For all the experiments, we use $\lambda_n = 10$ and for planning the iCEM optimizer (Pinneri et al., 2021). We use the same hyperparameters as Sukhija et al. (2024b)[11].

### D.5  Computational Costs

Table 3: Computation cost comparison for SOMBRL with different base algorithms.

| Algorithm | Training time |
| --- | --- |
| HUCRL (GPs) | 90 +/- 3 min (Pendulum), 31.5 +/ 2.5 min (MountainCar) |
| SOMBRL (GPs) | 30 +/- 0.6 min (Pendulum), 13.8 +/ 0.25 min (MountainCar) |
| MBPO-MEAN (Time per 100k steps, 1 ensemble, GPU: NVIDIA GeForce RTX 2080 Ti) | 9.6 +/- 0.2 min |
| MBPO-OPTIMISTIC (Time per 100k steps, 5 ensembles, GPU: NVIDIA GeForce RTX 2080 Ti) | 13.7 +/- 0.35 min |
| DREAMER (Time per 100k steps, GPU: NVIDIA GeForce RTX 4090) | 42.24 +/- 0.95 min |
| DREAMER-OPTIMISTIC (Time per 100k steps, 5 ensembles, GPU: NVIDIA GeForce RTX 4090) | 46.32 +/- 0.34 min |

---

[11]official implementation: https://github.com/lasgroup/opax

