# OpenReview forum: "SOMBRL: Scalable and Optimistic Model-Based RL"
_NeurIPS.cc/2025/Conference — NeurIPS 2025 poster_

### Official Review · Reviewer_dt6r · 2025-06-25

**Clarity:** 4
**Significance:** 3
**Originality:** 2
**Rating:** 5
**Confidence:** 4

**Summary:**

This paper proposes a simple optimism-in-the-face of uncertainty algorithm for model based RL. The algorithm works with any given uncertainty estimate; in theory, kernel estimates suffice whereas in practice, popular ensemble based methods suffice. Experiments are compelling on relatively simple environments.

**Questions:**

(1) How well do the authors suspect the method to perform in more complex tasks?
(2) Have the authors found the algorithm to be highly sensitive to the use of uncertainty estimates? What happens if the uncertainty estimate becomes unreliable?
(3) What would the authors regard as the most original (note: the distinction from originality and novelty) and most surprising aspects of their submission?

**Ethical Concerns:**

["NO or VERY MINOR ethics concerns only"]

**Final Justification:**

The authors pointed out a salient difference in between their result and Kakade et al, motivating the increased efficiency of their algorithm. This warranted me to raise my score.

**Limitations:**

Experimental domains are limited in their difficulty and sophistication. Consider, e.g. the RoboMimic, Libero, Calvin Kitchen control suites.

**Quality:**

3

**Strengths And Weaknesses:**

Strengths: the paper is very nicely written, theory is correct, well exposed, and relations to prior work are made clear. I apologize for not saying more here, but overall the execution is quite flawless and I do not have much more to add.

Weaknesses: though the strengths are apparent, I would say that there are two key weaknesses:

(1) Originality. Here, I would like to distinguish between originality and novelty. I believe the authors that their paper may be "novel" in the sense that no one has pursued this direction before. However, it does not feel particular original in the sense that anyone who is familiar with the ideas of Eluder dimension and OFU can believe that such an algorithm can be done and implemented. The salient difference between SOMBRL and the Kakade 2020 algorithm is that the uncertainty term is now aapplied on a per-time step basis, rather than aggregated across timesteps. This simplifies the optimism computation, but at the expense of some small statistical suboptimality. I don't think I find this particularly surprising, at least theoretically, nor would I be surprised if the authors of Kakade would have recommended this approach had someone tried to implement their algorithm theoretically. Ultimately, maybe the best conclusion here is "well, I guess somebody had to do it" and the authors did. And it was well executed. But still, I am not at all surprised (and in my own research, have been implementing something similar as a baseline). Again, I am not claiming this algorithm exists in the literature, but it feels more like "folklore principles" instantiated finally executed.

(2) I think the authors are taking liberties with the description "scalable." By the standards of modern RL, the settings are incredibly toy. The reason this is salient is that one need to be convinced that todays uncertainty-based methods are indeed robust enough for more dextrous, high-dimensional, or visually complex tasks. And because this method relies on explicit ensembling as a form of uncertainty estimation, the major research question is "to what extent to todays uncertainty estimates suffice to implement optimism." So far, I see the limited experimental domains as providing limited (though perhaps encouraging!) evidence.

---

> ### Author Rebuttal · Authors · 2025-07-28
>
> We thank the reviewer for their feedback and for acknowledging our writing, the simplicity of our algorithm, and the theoretical results.
>
> **Originality**: We agree that the algorithm is quite intuitive, and we find it to be one of its strengths. However, we think our theoretical analysis is original and not a straightforward extension of the results of prior work, such as Kakade et al 2020. Our analysis is based on a different question: how much exploration with the reward should I inject if I greedily plan with the mean dynamics? This underlying question is very different to what prior works like optimistic exploration methods consider. They pick the dynamics in the plausible set of learned models that are the most favourable. Hence, our approach only requires maximizing over the action/policy space, whereas the latter methods require optimizing over both the action space and the space of plausible dynamics. This is a much harder problem, which is practically less tractable and also does not scale with high input dimensions, e.g., visual control tasks. We show that through injecting exploration rewards, you can be optimistic. Which may sound intuitive, but we believe is an original theoretical insight. Our analysis requires us to study both the regret of the extrinsic reward and also bound the pure exploration objective (model uncertainty). Prior works have only studied them separately, i.e., Kakade et al 2020 only study the former, whereas OPAX (sukhija et al 2024) studies the latter.
> Moreover, regarding the reviewer’s comment: ``authors of Kakade would have recommended this approach had someone tried to implement their algorithm''. In fact, Kakade did recommend an empirical approach to their algorithm (see Section 3.1 on page 6) and their practical approach is based on a sampling-based approximation of the optimization over the dynamics. Moreover, they highlight the hardness of the optimization for their algorithm, and the method we chose is never discussed as a possible alternative. Furthermore, Curi et al (2020) propose the reparameterization trick instead to optimize over the dynamics. Compared to both these works, our much simpler and, compared to Curi et al (2020), our upper bound is exponentially better (see Theorem B.18).
> In addition, several works study uncertainty-based exploration empirically (Plan2Explore, Disagreement, Buisson-Fenet et al (2020)) and none of which provide any theoretical guarantee. Furthermore, other methods with theoretical guarantees, such as Mania et al. (2020); Wagenmaker et al. (2023); Treven et al. (2024) (see Section A for more), also propose a much more complex optimization problem instead of our simple objective. Moreover, since the seminal work from Abbasi-Yadkori et al 2011, most works have focused on optimistically selecting the dynamics model among a set of plausible dynamics. Having studied these prior works and worked on similar topics ourselves,  we believe our method is based on an original perspective and regret analysis that is different from prior works (as described above).
> In addition to the points above, most of these works focus on the finite horizon setting; we also provide our regret bound for the infinite horizon and the non-episodic case.
>
> **Scalability and Performance on Complex Tasks**: We evaluate our algorithm on well-established deep RL benchmarks, including high-dimensional tasks such as the humanoid (67-dimensional state and 21-dimensional action space), visual control problems, and also on a real-world hardware system. Across all our experiments, we found uncertainty-based exploration to either perform on par or better. We particularly build up on intrinsic exploration methods such as Plan2Explore and MaxInfoRL to scale our work for these settings. Hence, we believe this is encouraging evidence for testing our approach on more complex domains, and we find this a promising direction for future work.
>
> **Sensitivity to Uncertainty**: In our experiments, we found the use of ensembles as suggested in HUCRL, Plan2Explore, and MaxInfoRL to work quite robustly across tasks, i.e., we did not notice any sensitivity. However, these methods are also designed to be robust towards uncertainty estimation, e.g., Plan2Explore only quantifies uncertainty in a lower-dimensional latent space, and MaxInfoRL tunes how much exploration via uncertainty to inject into the system.  Nonetheless, theoretically, if the uncertainty estimates are off, e.g., too high, then the agent could end up overexploring, or, in the case of too low, underexploring. Hence, this can have an effect on the performance of the agent, and how to mitigate this, e.g., via better uncertainty quantification or auto-tuning mechanisms such as from MaxInfoRL, is an interesting research question.
>
> **Originality and Surprise**: As highlighted in our response above, we believe our perspective on regret and our theoretical analysis are not only novel but also original. Several prior works proposed no-regret model-based RL algorithms that require a much more complex and often intractable optimization over the set of dynamics models. These works start with the theoretical algorithm and then propose practical modifications, such as sampling-based optimization or the reparameterization trick, to solve the underlying optimization problem.
> We took a different approach by starting with a practical algorithm and showing that it has no regret. This required a different theoretical analysis from the prior work, as we highlight in our response above and the proof sketches in the paper. Furthermore, perhaps not surprising, but what we really like about our approach is that it provides a unified approach for MBRL, our algorithm can be used for pure (unsupervised) exploration by setting the extrinsic reward to 0, Curi et al 2020 and Kakade et al 2020 do not work for this setting and the OPAX algorithm which is used for pure exploration does not work in the setting with rewards. Furthermore, our approach also enjoys guarantees in the nonepisodic settings, which the other algorithms do not provide (in fact, Kakade et al, 2020, leave that as an open problem in Section 5).
>
>
> We hope that with our response, we could convince the reviewer of the strengths of our work, and we would appreciate it if they would consider increasing their score.

---

> ### Author Response · Authors · 2025-08-04
>
> Dear Reviewer,
>
> Thank you for your engagement in the review process. Since the discussion period is already half over, we would appreciate your response on our rebuttal. If there are any further questions, we are happy to answer them else we’d be glad if the reviewer would consider reevaluating their score.

---

> ### Comment · Area_Chair_xUBT · 2025-08-05
> **Please engage with the authors' response.**
>
> Dear reviewer dt6r,
>
> Thanks for your reviewing efforts so far. Please engage with the authors' response.
>
> Thanks,
> Your AC

---

> > ### Comment · Reviewer_dt6r · 2025-08-05
> > **Thank you for the feedback**
> >
> > Dear Authors,
> >
> > Thank you for the detailed response. While I still have doubts about the extent of the experiments (yes, humanoid is high dimensional, but Mujoco is widely regarded as a somewhat dated method). Still I agree that the analysis which considers planning through the mean dynamics with a bonus is certainly nicer than planning through the optimistimic MDP. I would say this is analoguous to the style of the Jin et al Linear MDP paper, which leads to an efficient algorithm. While this is a nice observation, I still don't consider the proof techniques to be particularly novel by the standards of a theoretical contribution. Altogether, however, I appreciate the clarification and will raise my score slightly.

---

> > > ### Author Response · Authors · 2025-08-05
> > >
> > > Dear Reviewer,
> > >
> > > Thanks for your active engagement and feedback. We are happy to see you acknowledge our analysis and appreciate your positive reevaluation of our work.
> > > Furthermore, we recognize your concerns regarding MuJoCo; however, we also think the robotics community has recently shown impressive real-world results powered by the MuJoCo physics engine (1, 2). Additionally, besides testing our method in simulation, we also show real-world online RL results on the RC car.
> > >
> > > Thanks again for your engagement in reviewing our work.
> > >
> > > **References**:
> > >
> > > 1. https://arxiv.org/abs/2502.08844
> > >
> > > 2. https://arxiv.org/abs/2410.03654

---

### Official Review · Reviewer_ZcLd · 2025-06-28

**Clarity:** 2
**Significance:** 3
**Originality:** 2
**Rating:** 4
**Confidence:** 3

**Summary:**

The authors proposed SOMBRL (Scalable an Optimistic MBRL) approach based on the pricipl of optimism in the face of uncertainty. SOMBRL learns an uncerrtainty-aware model of the true dynamics and use it to find a policy maximizig te extrinsic reward and moddel uncertainty to stimulate the exploration. Furthermore, the authors provide thoretical sublinear regreet results,

**Questions:**

1. To avoid potential misunderstandings, I would like to clarify the following: In Equation (8), the formulation appears different from MBPO. When applying your method to MBPO, do you mean that the synthetic data generation still uses only the mean dynamics (unlike the original MBPO), and that uncertainty is incorporated in the rewards of both real and synthetic data used for training the Q-networks and the actor network?

2. In model-based RL, I think that the predictive performance of the model is crucial in practice. Introducing additional rewards after rollout might lead to overestimation in the synthetic data. I am concerned this could actually degrade performance rather than improve it. I wonder what your thoughts are on this point, or if you have already addressed this issue in the paper, please let me know.

3. The figures seem to have inconsistent line widths, which affects visual consistency. In Figure 3, for example, if this is indeed model-based policy optimization, the label on the left should be "Dreamer." Personally, I feel consistency across figures would improve the presentation. Additionally, I do not quite understand the purpose of Figure 1.

**Ethical Concerns:**

["NO or VERY MINOR ethics concerns only"]

**Final Justification:**

While existing uncertainty-based exploration methods generally lack theoretical guarantees, **this work presents sublinear regret bounds for a practical exploration strategy.** Moreover, the validity of the algorithm has been demonstrated across various settings, including both continuous and discrete actions. However, there are **some mathematical notation** that need to be revised, and **it is regrettable that no comparison was made with existing exploration methods.**

Therefore, I recommend **a weak accept.**

**Limitations:**

yes

**Paper Formatting Concerns:**

I could not find any issue.

**Quality:**

3

**Strengths And Weaknesses:**

Strength
1. Theoretical Justification
2. Simple and Scalable Methods
3. High Compatibillity

Weaknesss
1. The figures and descriptions of the experiments are overly cluttered and lack clarity.
2. There are some model-base RL method considering uncertainty based on the model for dynamics additionally.

---

> ### Author Rebuttal · Authors · 2025-07-28
>
> We thank the reviewer for their feedback and for acknowledging the simplicity, theory, scalability, and flexibility of our method.
>
> **Figures**:  In Figure 3, we have 2 panels. The left panel shows our experiments with MBPO, and the right with Dreamer.  We will clarify the different algorithms in our figure further and also use consistent line widths across the figures. Figure 1 is used as a teaser to showcase the diverse settings we consider in our experiments. Would the reviewer like us to remove the figure? If yes, we are happy to do so.
>
> **Other works on model uncertainty-based exploration**: We agree that other works also propose using the model uncertainty as a reward for exploration. In fact, we cite several works in Section A of the paper. However, to the best of our knowledge, we are the first to show that this simple approach enjoys theoretical grounding for most common RL settings under the same assumptions as prior principled exploration works (that are much more complex and not scalable), such as (1-5). We think this is an important result since it provides theoretical grounding for a much simpler and intuitive exploration strategy. To establish this result, we required a novel analysis as opposed to the prior works, which we tried to highlight in our proof sketch.
>
> **Planning in MBPO and how model uncertainty improves performance**: We use the mean model to use model-generated rollouts. We add the epistemic uncertainty of our model to the extrinsic reward (both for real and synthetic data) and use it for training the policy and critic. We find this to work quite robustly across our experiments. Moreover, by adding the uncertainty as a reward, we encourage the policy to explore areas of the state space where our model has high uncertainty. This typically corresponds to areas where we have less data/high model error. Thereby, by collecting data in high uncertainty regions, we improve our model. Note that uncertainty-based sampling is a well-established exploration technique (see the seminal work from Lewis & Catlett, 1994), even for RL (6-8).
> Intuitively, by adding the model uncertainty to the reward, we improve our model, which then leads to better policy optimization. Therefore, we believe that adding the reward does not lead to overestimation; instead, it improves our model and thereby results in better planning performance. The extreme case of our method is studied by works such as (6, 7), where only the model uncertainty is used as reward, and they show that this results in strong planning with a learned model for multiple downstream tasks.
>
>
> We hope our response clarifies the reviewers' concerns. In summary, we provide theoretical grounding for a practical RL algorithm. Even though other works have also proposed using the model uncertainty as an intrinsic reward, to the best of our knowledge, we are the first to show that this has no regret for several common RL settings. Next, we also evaluate the exploration strategy across different tasks; state-based (including the high-dimensional humanoid tasks), visual control, and on real-world hardware.
>  We think this makes our paper a strong contribution and would appreciate it if the reviewer would increase their score.
>
> **References**:
> 1. https://proceedings.neurips.cc/paper/2020/hash/aee5620fa0432e528275b8668581d9a8-Abstract.html
>
> 2. https://proceedings.neurips.cc/paper/2020/hash/a36b598abb934e4528412e5a2127b931-Abstract.html
>
> 3. https://www.jmlr.org/papers/v23/20-807.html
>
> 4. https://proceedings.neurips.cc/paper_files/paper/2023/hash/31e018f43ab9c7065c058cc2c5848128-Abstract-Conference.html
>
> 5. https://proceedings.neurips.cc/paper_files/paper/2023/hash/836012122f3de08aeeae67369b087964-Abstract-Conference.html
>
> 6. https://proceedings.mlr.press/v119/sekar20a.html
>
> 7. https://proceedings.neurips.cc/paper_files/paper/2023/hash/77b5aaf2826c95c98e5eb4ab830073de-Abstract-Conference.html
>
> 8. https://proceedings.mlr.press/v97/pathak19a.html

---

> > ### Comment · Reviewer_ZcLd · 2025-08-02
> >
> > Thank you for your response.
> >
> > In the AI community, there are many results that are empirically effective but not yet well understood theoretically. I believe that the sublinear effect of adding exploration provides a meaningful justification for such phenomena in model-based reinforcement learning.
> >
> > **Q1**: I have an additional question: In your theorems, the right-hand sides of the inequalities do not include any explicit dependence on $\delta$. In my understanding, such bounds typically involve terms like $\log(1/\delta)$, which may be hidden. Am I correct? If so, I’m concerned that the current form of the theorem might lead to misunderstanding. Perhaps it would be clearer to use the $\widetilde{O}$ notation to make this implicit dependence more transparent. But if my understanding is mistaken, I would appreciate your clarification.

---

> > > ### Author Response · Authors · 2025-08-02
> > >
> > > We thank the reviewer for their active engagement and for acknowledging our theoretical results.
> > >
> > > **Dependence on $\delta$**: Indeed, there is a  $\log(1/\delta)$ dependence in the regret bound (see Theorem 2 in https://arxiv.org/pdf/1704.00445). We focused the O-notation only on the dependence with respect to $N$, but we will adapt the bound according to what the reviewer suggested to avoid misunderstandings. Thanks for your feedback on this!
> > >
> > > We appreciate the reviewer’s feedback and hope we could address their concerns.

---

> ### Author Response · Authors · 2025-08-04
>
> Dear Reviewer,
>
> Thank you for your engagement in the review process. If there are any further questions, we are happy to answer them else we’d appreciate it if the you would consider reevaluating their score.

---

### Official Review · Reviewer_ST8B · 2025-07-02

**Clarity:** 3
**Significance:** 3
**Originality:** 3
**Rating:** 4
**Confidence:** 3

**Summary:**

This paper presents SOMBRL, a scalable and optimistic model-based reinforcement learning algorithm that combines extrinsic rewards with epistemic uncertainty to guide exploration. The key idea is to use a weighted sum of the reward and the model uncertainty during planning. The method is simple yet theoretically grounded, achieving reasonably good results across lots of RL settings. This work shows strong empirical performance in both simulation (state-based and visual control tasks) and a real-world RC car setting.

**Questions:**

- What is the key intuition for why maximizing epistemic uncertainty improves convergence and stability? Can this be interpreted as a kind of regularization?
- How is the uncertainty model accuracy affect the results?

**Ethical Concerns:**

["NO or VERY MINOR ethics concerns only"]

**Final Justification:**

No change

**Limitations:**

- The paper briefly mentions the computational overhead of learning a probabilistic model but does not thoroughly discuss the accuracy of uncertainty estimation and its potential impact.

**Quality:**

2

**Strengths And Weaknesses:**

Strengths:
- This is an important problem. Exploration in model-based RL is a key challenge for real-world deployment of RL systems.
- The solution is simple but clear. This work proposes to learn an uncertainty model along with the environment model.
- This work shows strong empirical results. The method shows consistent performance improvements across various high-dimensional benchmarks.

Weaknesses
- The abstract suggests that SOMBRL targets realistic settings, but the theoretical results still rely on continuity assumptions. While common in literature, these assumptions may not always hold in complex real-world environments.
- The guarantees rely on well-calibrated uncertainty models, but how accurate these uncertainty models is not explored. Further, the intuition of why this method works was not clearly stated.

---

> ### Author Rebuttal · Authors · 2025-07-28
>
> Thanks for your review, for highlighting the simplicity of our method and our theoretical as well as empirical results.
>
> **Tackling realistic settings**: We agree with the reviewer that certain continuity assumptions are required for theoretically studying nonlinear systems in continuous spaces. However, we try to bridge the gap between theory and practice by thoroughly evaluating our method across different real-world settings, such as state-based control, visual control, and even real-world hardware experiments. We believe that through this, we could illustrate the strengths of our algorithms in real-world settings. We will try to clarify this further in the abstract, but we are also happy to receive any further suggestions from the reviewer on this matter.
>
> **Intuition of uncertainty-based exploration**: Uncertainty-based exploration is a well-established exploration strategy (see Lewis & Catlett, 1994). Intuitively, by rewarding the agent when going to areas with high uncertainty, we encourage the agent to collect data where it has the most information to gain, i.e., reduce its uncertainty about the system the most.
> Therefore, as the agent continues learning, it explores all relevant regions where the uncertainty is high and reduces its epistemic uncertainty of the dynamics.  Here, relevance is determined by the extrinsic reward of the state action space. Under the well calibration assumption, this implies that our learned model converges to the true dynamics. In contrast, greedy algorithms do not ensure the necessary coverage of the state-action space, i.e., exploration, to ensure convergence, and random exploration strategies are not directed. We actively visit points where we have less knowledge and learn, whereas greedy and random exploration approaches often tend to either visit the same points, under-explore, and get stuck in a local optimum (greedy) or over-explore the areas that don't carry much information about the unknown function (random exploration).
>
> **Effects of uncertainty model accuracy**:  In our experiments, we found the use of ensembles as suggested in HUCRL, Plan2Explore, and MaxInfoRL to work quite robustly across tasks, i.e., we did not notice any negative effects. However, these methods are also designed to be robust towards uncertainty estimation, e.g., Plan2Explore only quantifies uncertainty in a lower-dimensional latent space, and MaxInfoRL tunes how much exploration via uncertainty to inject into the system.  Nonetheless, theoretically, if the uncertainty estimates are off, e.g., too high, then the agent could end up overexploring, or, in the case of too low, underexploring. Hence, this can have an effect on the performance of the agent, and how to mitigate this, e.g., via better uncertainty quantification or auto-tuning mechanisms such as from MaxInfoRL, is an interesting research question.
>
>
> SOMBRL is a unique algorithm since it offers both simplicity and theoretical grounding for many common RL settings. Across our experiments, we show that SOMBRL performs strongly in real-world settings. We hope our response addresses your concerns, and we would appreciate it if you would increase our score.
>
>
>
> **References**:
>
> 1. https://proceedings.mlr.press/v119/sekar20a.html
>
> 2. https://proceedings.neurips.cc/paper_files/paper/2023/hash/77b5aaf2826c95c98e5eb4ab830073de-Abstract-Conference.html
>
> 3. https://proceedings.mlr.press/v97/pathak19a.html
>
> 4. https://proceedings.neurips.cc/paper/2020/hash/a36b598abb934e4528412e5a2127b931-Abstract.html

---

> ### Author Response · Authors · 2025-08-04
>
> Dear Reviewer,
>
> Thank you for your engagement in the review process. Since the discussion period is already half over, we would appreciate your response on our rebuttal. If there are any further questions, we are happy to answer them else we’d be glad if the reviewer would consider reevaluating their score.

---

> ### Comment · Area_Chair_xUBT · 2025-08-05
> **Please engage with the authors' response.**
>
> Dear reviewer ST8B,
>
> Thanks for your reviewing efforts so far. Please engage with the authors' response.
>
> Thanks,
> Your AC

---

> > ### Author Response · Authors · 2025-08-07
> >
> > Dear Reviewer,
> >
> > The discussion period deadline is nearing. We believe our response above addresses your questions/concerns and we would like your feedback on the rebuttal.
> > If your concerns are addressed, we would appreciate it if you would reevaluate our score.
> >
> > As highlighted in our response above and also acknowledged by other reviewers, our work addresses an important gap for exploration in MBRL by proposing a simple and scalable/tractable alternative for principled/theoretically grounded exploration. We think this makes for a strong and important contribution to RL research.
> >
> > Thanks for your support in reviewing our work.

---

### Official Review · Reviewer_2BDK · 2025-07-02

**Clarity:** 3
**Significance:** 2
**Originality:** 1
**Rating:** 2
**Confidence:** 4

**Summary:**

The paper proposes a new model-based RL algorithm that incorporates systematic exploration through intrinsic rewards designed to drive the agent to more uncertain regions of the state space. The authors show that under certain regularity conditions, the proposed algorithm achieves a sublinear regret while being simpler and more scalable that previous contendors. Finally, the authors provide some experimental results on some RL benchmarks to validate their approach.

**Questions:**

Can you examples of setting where Assumption 5.2 holds?

**Ethical Concerns:**

["NO or VERY MINOR ethics concerns only"]

**Limitations:**

yes

**Paper Formatting Concerns:**

No concerns as far as I could tell.

**Quality:**

2

**Strengths And Weaknesses:**

Strenghts:
- It is nice that the paper provides explicit theoretical guarantees for the proposed algorithm under the regularity conditions considered. It is a bonus that the guarantees extend to the non-episodic settings.
- The paper is relatively well written and easy to follow.

Weaknesses:
- First, the authors claim that they are the first to provide guarantees for RL with continuous state and action spaces. This inaccurate (at least not for the model-based setting); see The Statistical Complexity of Interactive Decision Making by Foster et al 2023.
- The guarantees provided require strong assumptions on the model. This is not really highlighted as a limitation in the paper.
- Including a measure of epistemic uncertainty as a bonus term to drive exploration is not new; see for example the paper Exploration by Random Network Distillation by Burda et al. 2018. So I am not really seeing the novelty here.
- In the theory section, epistemic uncertainty is measure with the help of a kernel. In the experiment, the authors use an ensemble model to measure the uncertainty. This feels like testing a different algorithm to me.
- The literature survey and comparison with existing work isn't exhaustive enough. The two papers above are a good example, and maybe also the paper Sample Efficient Deep Reinforcement Learning via Local Planning by Yin et al. 2023.
- The experimental results aren't convincing as far as exploration goes. For example, the score on Montezuma's revenge caps at 2500, which is less the model-free RL algorithms in the last two paper mentioned above. So this is does not make a good case for the model based approach of the paper.

---

> ### Author Rebuttal · Authors · 2025-07-28
>
> Thank you for your review and for acknowledging our theoretical contribution. We would like to highlight that we do not propose a single algorithm; instead, we theoretically study the framework of combining extrinsic rewards with the model epistemic uncertainty. We add it on top of existing MBRL algorithms such as MBPO, Dreamer and SimFSVGD and show that besides enjoying theoretical grounding, it also yields strong performance. As the reviewer highlighted, people have proposed using the model epistemic uncertainty in the past, e.g., Plan2Explore with Dreamer, RND with PPO, OPAX with model-based RL and MPC, and more (see Section A for more works). However, we are the first to show that this approach has sublinear regret, under the same assumptions as other principled exploration methods such as Kakade et al 2020, Curi et al 2020, and more. As opposed to these works, our approach is much more practical than the aforementioned prior works, and we demonstrate this in the experiments.
>
> **Claim on first to give theoretical guarantees**: We are sorry for any confusion caused. We do not claim to be the first to give theoretical guarantees for model-based RL in continuous state action spaces. In fact, in Table 1, we highlight other works that, under the same assumptions as ours, give theoretical guarantees for the different RL settings. Moreover, we have more detailed related works in Section A, where we also discuss other works that theoretically study model-based RL. We will add Foster et al 2023 to this section. Due to the nine-page limit, we only kept a smaller related work discussion in the main paper. Finally, our central claim is that we are the first to show that leveraging model uncertainty estimates as intrinsic rewards gives sublinear regret for most common RL settings in MBRL (finite horizon, discounted horizon, non-episodic, and unsupervised RL). We believe this is a strong contribution, since it provides theoretical grounding for a much simpler approach for principled exploration in RL.
>
> **Assumptions**: We would like to highlight that there are several works that study MBRL theoretically under the same assumption as ours  (Kakade et al. (2020); Curi et al. (2020); Mania et al. (2020); Wagenmaker et al. (2023); Treven et al. (2024), or Section A). Moreover, the kernelized assumption is also ubiquitous in Bandits with continuous spaces (GPUCB, IDS, TS). Hence, to the best of our knowledge, these assumptions are quite common when studying online decision-making algorithms in continuous spaces. Practically, the RKHS assumption captures any function that can be represented with $f(s, a) = \phi(s, a) \theta$, where $\phi$ is a nonlinear, potentially infinite-dimensional feature map (see Section 1 in Kakade et al 2020, for a discussion on the generality of this assumption). Kernelized models are commonly used for learning and control in the real world (c.f., 1-6).
> Extending these results to deep learning models such as RSSMs and BNNs is still an open research question, even for Bandits (there is some research in this line with NTKs (7)). We will highlight this in the limitations section.
>
> **RND and other intrinsic exploration methods**: In our related works (Section A), we discuss RND and other intrinsic exploration methods. As highlighted above, we are the first to show that combining extrinsic rewards with the model epistemic uncertainty gives sublinear regret for several RL settings. None of the discussed works provides such theoretical grounding, and as compared to the methods that do have theoretical results, our approach is much simpler, as highlighted by the reviewer. In summary, our key novelty is to show for several RL settings that adding model epistemic uncertainty is not only empirically strong but also theoretically backed.
> This requires a novel analysis as highlighted in our proof sketches in Section 5.
>
> **Kernelized setting and BNNs**: In Figure 2 (Section 5), we report experiments for the kernelized case for both finite-horizon and non-episodic RL settings. In the experiments, we evaluate our approach with SOTA deep learning models such as BNNs and RSSMs, where the well-calibration is not guaranteed as in the kernelized case. Crucially, our exploration strategy of adding epistemic uncertainty to the extrinsic reward is agnostic to what model, planner, or in general, base algorithm we use, e.g., MBPO, Dreamer, or SimFSVGD. However, we empirically show that across all experiments, adding the uncertainty reward helps the agent achieve on par or better performance. Particularly, in our hardware experiments, we illustrate the benefits of the uncertainty bonus in the real world, where the agent fails to explore meaningfully without the additional exploration reward.
>
> **Literature Review**: As discussed in our response above, we do have a more detailed related works section in Section A. We will add Foster et al 2023, and Yin et al 2023 to it. Foster et al 2023 propose a theoretical algorithm, which also requires solving a min-max optimization problem that is intractable in practice. Our RKHS assumption is the same as the reliazbility assumption in Foster et al 2023, where the model class is defined by the RKHS.
> Yin et al 2023 empirically study uncertainty-based exploration methods for the setting where the environment can be reset to previously visited states. In comparison, our method is both practical and theoretically grounded for several common RL settings.
>
> **Experiments**: We add our exploration bonus on top of several base RL algorithms (MBPO, Dreamer, SimFSVGD). In the case of Montezuma, we use Dreamer, a well-established and SOTA RL algorithm for sample-efficiency and performance. We believe the performance of our algorithm is affected by the base algorithm itself, i.e., the expressivity of the RSSM model and the planner. Generally, we find that adding our exploration objective helps in obtaining better performance across the different base algorithms we try (see Figure 3 and 4). However, other elements, such as how to learn a model, how to obtain a policy from the learned models, are also important in RL but not the focus of our work.
>
> **Examples where Assumption 5.2 holds**: See discussion on assumptions.
>
>
> We hope our response clarifies the concerns raised. We believe that our work addresses an important gap for exploration in MBRL by proposing a simple and scalable/tractable alternative for principled exploration. Our theoretical analysis is novel and applied to several important RL settings. In addition, we evaluate our work on state-based, visual control, and real hardware experiments. We think this makes our work original and a strong contribution to RL research. We hope we could bring our opinion across better to the reviewer, and would appreciate it if they would increase our score.
>
> **Additional References**:
>
> 1. https://ieeexplore.ieee.org/document/8754713
>
> 2. https://ieeexplore.ieee.org/abstract/document/8467518?casa_token=zXxHSxz19hYAAAAA:CZc9OA6MklYZrZ-VNrMK_XvBxeVpCbWAStnYeVdChbyVvU6JVQsfyMXzmXAu6cGknKK73bKNqpc
>
> 3. https://arxiv.org/abs/2103.04490
>
> 4. https://proceedings.mlr.press/v139/song21b.html
>
> 5. https://proceedings.mlr.press/v144/boffi21a.html
>
> 6. https://www.science.org/doi/full/10.1126/scirobotics.abm6597
>
> 7. https://arxiv.org/abs/1806.07572

---

> ### Author Response · Authors · 2025-08-04
>
> Dear Reviewer,
>
> Thank you for your engagement in the review process. Since the discussion period is already half over, we would appreciate your response on our rebuttal. If there are any further questions, we are happy to answer them else we’d be glad if the reviewer would consider reevaluating their score.

---

> ### Comment · Area_Chair_xUBT · 2025-08-05
> **Please engage with the authors' response.**
>
> Dear reviewer 2BDK,
>
> Thanks for your reviewing efforts so far. Please engage with the authors' response.
>
> Thanks,
> Your AC

---

> > ### Author Response · Authors · 2025-08-07
> >
> > Dear Reviewer,
> >
> > The discussion period deadline is nearing. We believe our response above addresses your questions/concerns and we would like your feedback on the rebuttal.
> > If your concerns are addressed, we would appreciate it if you would reevaluate our score.
> >
> > As highlighted in our response above and also acknowledged by other reviewers, our work addresses an important gap for exploration in MBRL by proposing a simple and scalable/tractable alternative for principled/theoretically grounded exploration. We think this makes for a strong and important contribution to RL research.
> >
> > Thanks for your support in reviewing our work.

---

> > > ### Comment · Reviewer_2BDK · 2025-08-08
> > > **Thanks for the detailed reply**
> > >
> > > Looking at your rebuttal and the paper again, I unfortunately still feel that:
> > > 1- The theoretical results aren't strong enough: kernelized assumption is strong and not reflective of real environements.
> > > 2- The empirical results aren't convincing enough as far as improving on exploration goes.
> > > I will maintain my score.

---

> ### Author Response · Authors · 2025-08-09
>
> While we respect your perspective, your statements contradict the evidence in our paper. Below we reiterate this evidence, already presented in our rebuttal and the paper, by directly addressing the remaining concerns.
>
> 1. We were responding to your claim that our assumption is "not reflective of real environments." As noted in the rebuttal, our assumptions are in line with numerous prior works in this area. While no assumption captures the full complexity of real environments, we show that our algorithm improves upon models such as RSSMs, which are widely recognized to be among the most competitive baselines for real-world RL on autonomous systems [1] [2]. **We further demonstrate that our algorithm outperforms the baseline in a real-world RC learning testbed.** Motivated by our theoretical analysis, these results repeatedly show compatibility with real-world scenarios.
>
> 2. Here we are responding to the claim that better performance is "not fully reflected in the results shown in the current version of the paper." Among several others, we would like to point out Fig. 3 and Fig. 4 in the paper, where SOMBRL is either on par or outperforms the baseline method. Our claim is straightforward: **baseline_method + SOMBRL >= baseline_method**, with only a minimal increase in computational cost (see Table 3 in the paper).
>
> [1] https://arxiv.org/abs/2206.14176
>
> [2] https://arxiv.org/abs/2312.09906

---

### Note · Authors · 2025-08-12

Dear AC,

Below we provide a brief summary of the discussions from the rebuttal period.

**We show that intrinsic exploration via model epistemic uncertainty has sub-linear regret for several common RL settings**: Exploration through model disagreement/uncertainty has been empirically studied by prior works but theoretically it is much less understood. In contrast, theoretical grounded methods provide strong convergence guarantees but are practically intractable for many real-world settings.
We show that intrinsic exploration via model epistemic uncertainty gives sublinear regret under many common RL settings. Thereby providing theoretical grounding for a practical and scalable exploration algorithm. We were happy to see reviewers also acknowledge this finding.

**Assumptions**: We study a general nonlinear system in continuous state and action spaces. For our theoretical results, we make continuity assumptions on the system that are common in prior work and also used for real-world applications (see our response to Reviewer 2BDK). Providing theoretical guarantees without any regularity assumptions on the underlying system is an ill-posed problem and to the best of our knowledge our assumptions are very general/state-of-the-art for studying nonlinear dynamics.

**Experiments**:  We evaluate our method in simulation on state-based and visual-control tasks as well as in the real-world on a RC car hardware platform. We combine SOMBRL with several base MBRL methods, e.g., Dreamer, MBPO etc., and show that it performs better or at least on par to the base methods. In our hardware experiment, we show that SOMBRL is able to solve the task in only 20 episodes, i.e., demonstrating remarkable sample efficiency (model-free on-policy methods such as RND mentioned by reviewer 2BDK require orders of magnitude more samples).

We were happy to see the active engagement of reviewers ZcLd and dt6r in the rebuttal period and thank them for their time and valuable feedback of our work. Unfortunately, we could not engage in a discussion with the remaining reviewers within the rebuttal deadline. However, we hope our message above together with the rebuttal clarifies and underlines the strengths of our work.

We think our work is an important contribution to RL research since it provides theoretical grounding for practical exploration strategies that scale to high-dimensional as well as real-world settings as we demonstrate in our results.

Thanks,

Authors

---

### Decision · Program_Chairs · 2025-09-17

**Decision:**

Accept (poster)

**Comment:**

This paper presents a model-based RL algorithm that uses an estimate of the epistemic uncertainty of the dynamics model to perform explorative planning during learning. Theoretical analysis studies the performance of the algorithm and shows it has sublinear-in-N regret in continuous state and action spaces. Experiments demonstrate the efficacy of the approach when applied to existing MBRL approaches in state-based and visual settings.

Reviewers mentioned concerns about novelty, specifically prior theoretical and empirical work has proposed using epistemic uncertainties in model-based RL. Quoting from a reviewer: "Ultimately, maybe the best conclusion here is "well, I guess somebody had to do it" and the authors did. And it was well executed. But still, I am not at all surprised (and in my own research, have been implementing something similar as a baseline). Again, I am not claiming this algorithm exists in the literature, but it feels more like "folklore principles" instantiated finally executed. "

While it does appear there's a disconnect between the theoretical results and the algorithm tested, and that the algorithm tested bears similarity to prior theoretical work, the simplicity of the method combined with the clear empirical gains suggest to me sufficient novelty warranting publication. The characterization of the approach being an instance of "folklore principles finally executed" is another compelling argument for acceptance, in the interest of turning folklore into a more well-known result that future work can more easily build off of.